# What Makes a Desired Graph for Relational Deep Learning?

**Yao Cheng** [1]  **Siqiang Luo** [1]

## Abstract

Relational deep learning (RDL) converts relational databases (RDBs) into heterogeneous graphs, but graphs derived directly from database schemas are often not well suited for how graph neural networks (GNNs) perform relational reasoning. We study what makes a relational graph suitable for deep learning and show that schema-derived graphs suffer from two systematic failures: information overload and semantic fragmentation. Our empirical analysis reveals that the desired graph is not the raw schema, but a result of controlled structural adaptation. Performance depends on balancing two operations: mitigating information overload via filtering, and repairing semantic fragmentation via injection. Specifically, filtering serves as a bias-variance knob with non-monotonic effects, while injection improves performance only when it explicitly restores the relational dependencies missing from the original schema. Based on these findings, we develop an end-to-end structural optimizer that applies both operations to adapt relational graphs automatically. Across 26 tasks spanning classification, regression, and recommendation, the optimized graphs consistently improve accuracy while often reducing inference cost. Code and data are available at https://github.com/cy623/Structural_Optimizer_RDL.git.

## 1. Introduction

Relational databases (RDBs) (Codd, 1970; Ke et al., 2017; Banachewicz & Massaron, 2022) form the backbone of modern data infrastructure. They are organized under normalization principles (Fagin, 1979; Maier, 1983) to ensure data integrity and storage efficiency. Recently, Relational Deep Learning (RDL) (Fey et al., 2024) has made it possible to apply Graph Neural Networks (GNNs) (Schlichtkrull et al., 2018; Liu et al., 2026c;b;a) directly to RDBs by representing tables and foreign-key relations as heterogeneous graphs.

While intuitive and convenient, this conversion introduces a fundamental mismatch in design objectives. RDB schemas are optimized for reliable storage and efficient querying, which often requires decomposing information across multiple normalized tables based on various normal forms (e.g., 2NF, 3NF or BCNF) to remove different levels of data anomalies. In contrast, GNNs rely on local connectivity and short relational paths to propagate information efficiently.

Consequently, a graph that is derived from a database schema in a standardized way is rarely desired for learning (Chen et al., 2025; Dwivedi et al., 2025). Such graphs typically exhibit two systematic failures. First, *information overload*: many attributes and entity types in an RDB are weakly or not at all related to the prediction target (e.g., identifiers, logging fields). When mapped into a graph, these variables increase representational complexity without adding useful predictive signal. Second, *semantic fragmentation*: normalization decomposes meaningful relationships into chains of foreign-key links. For example, a "User–Item" interaction is often represented as "User–Order–OrderItem–Item". This forces message passing to traverse long paths to recover a dependency that is conceptually direct, increasing both computational depth and the risk of signal decay. These effects imply that a schema-derived graph is not necessarily a task-faithful graph: preserving all tables and relations may maintain database semantics, but it often obscures the relational structure most relevant for prediction.

In this paper, we move beyond model architecture design and focus on a more fundamental problem: *what makes a desired graph for Relational Deep Learning?* We argue that the desired graph is not the raw database schema, but a task-dependent transformation of it. Constructing a desired graph requires two complementary operations. On one hand, we must remove relational structures and attributes that contribute little to the prediction target. On the other hand, we must introduce relational links that directly expose the dependencies required for reasoning. In this sense, graph construction is not a forward mapping from schema to topology, but a controlled process of compression and

[1]Nanyang Technological University, College of Computing and Data Science, Singapore. Correspondence to: Siqiang Luo <siqiang.luo@ntu.edu.sg>.

*Proceedings of the 43rd International Conference on Machine Learning*, Seoul, South Korea. PMLR 306, 2026. Copyright 2026 by the author(s).

augmentation guided by the learning objective.

Accordingly, this paper adopts a principled two-stage approach. We first investigate the determinants of an effective relational graph by systematically analyzing how filtering and structural injection influence predictive performance across a diverse array of real-world databases. This analysis reveals consistent, task-dependent regularities governing the trade-off between mitigating information overload and restoring relational connectivity. Guided by these insights, we design a structural optimizer that operationalizes these two operations within an end-to-end trainable framework. Extensive evaluation on 23 tasks from the RELBench benchmark (Robinson et al., 2024) demonstrates that our learned graphs consistently outperform those derived from raw schemas and existing relational learning pipelines. The main contributions of this work are summarized as follows:

- We provide a systematic analysis of how graph structure derived from relational databases affects relational deep learning, identifying *information overload* and *semantic fragmentation* as the main sources of performance degradation.

- We reveal that constructing effective graphs requires a delicate trade-off between filtering out structural noise and injecting task-relevant signals. Our empirical analysis further characterizes how this structural balance is dictated by the semantics of the downstream task.

- We translate these insights into an end-to-end structural optimizer that automatically adapts relational graphs to the learning objective, and demonstrate its effectiveness on a large benchmark of real-world relational databases.

## 2. Preliminaries

### 2.1. Relational Database

A relational database $\mathcal{D}$ is defined as a tuple $\mathcal{D} = (\mathcal{T}, \mathcal{R}_{\mathrm{db}})$, where $\mathcal{T} = \{T_1, \ldots, T_N\}$ is a finite set of $N$ tables, and $\mathcal{R}_{\mathrm{db}} \subseteq \mathcal{T} \times \mathcal{T}$ is a set of referential constraints. Each table $T_i \in \mathcal{T}$ has a set of attributes $A_i$, and each constraint $(T_i, T_j) \in \mathcal{R}_{\mathrm{db}}$ indicates that $T_i$ contains a foreign key referencing the primary key of $T_j$. The database schema $(\mathcal{T}, \mathcal{R}_{\mathrm{db}})$ defines the organizational structure of entities and their relationships, optimized for data integrity and efficient query processing rather than representation learning.

### 2.2. RDB-derived Heterogeneous Graph

A heterogeneous graph is defined as $\mathcal{G} = (\mathcal{V}, \mathcal{E}, \phi, \psi)$, where $\mathcal{V}$ denotes the set of nodes and $\mathcal{E} \subseteq \mathcal{V} \times \mathcal{V}$ denotes the set of directed edges. The function $\phi : \mathcal{V} \to \mathcal{N}$ maps each node to a node type in the set $\mathcal{N}$, while $\psi : \mathcal{E} \to \mathcal{R}$ maps each edge to a relation type in the set $\mathcal{R}$. In RDB schema-derived graphs, node types correspond to database tables,

i.e., $\mathcal{N} = \mathcal{T}$. The conversion from a relational database $\mathcal{D} = (\mathcal{T}, \mathcal{R}_{\mathrm{db}})$ to a heterogeneous graph $\mathcal{G}$ proceeds as follows. First, each tuple in a table $T_i \in \mathcal{T}$ is mapped to a node $v \in \mathcal{V}$ with node type $\phi(v) = T_i$ and feature vector $x_v$ derived from its attributes. Second, for each referential constraint $(T_i, T_j) \in \mathcal{R}_{\mathrm{db}}$, directed edges are created between corresponding nodes, with edge types $\psi(e) \in \mathcal{R}$ reflecting the underlying foreign-key relationships. Edge directions follow the referential semantics from foreign-key tables to referenced tables.

To support supervised learning, each predictive task introduces a dedicated training table $T_{\mathrm{train}}$, which contains labels but no input features. During training, seed nodes are sampled from $T_{\mathrm{train}}$. Each training node $v \in T_{\mathrm{train}}$ is associated with a timestamp $t_v$, a ground-truth label $y_v$, and foreign-key links connecting it to relevant entity tables. Temporal information is essential, for example, when predicting whether a user will purchase an item at time $t$, the model should rely solely on data observed prior to $t$; otherwise, temporal leakage will occur. To ensure causal consistency in downstream learning tasks, we construct time-consistent subgraphs during sampling. For a given seed node at time $t$, the temporal neighborhood includes only nodes whose timestamps satisfy $t_v \leq t$, thereby excluding any future information and preventing leakage.

### 2.3. Graph Neural Networks

A GNN learns node or edge representations by recursively aggregating information from local neighborhoods. When applied to heterogeneous graphs, the aggregation and transformation functions must account for the diversity of node and relation types. Each node $v \in \mathcal{V}$ is associated with an initial feature vector $x_v$ derived from the corresponding tuple in the relational database. These features are linearly transformed to initialize the hidden state using learnable parameters $W_{\phi(v)}$:

$$h_v^{(0)} = W_{\phi(v)} x_v, \tag{1}$$

which serves as the input representation for message passing in GNNs. Formally, for a heterogeneous graph $\mathcal{G} = (\mathcal{V}, \mathcal{E}, \phi, \psi)$, the message-passing process of a heterogeneous GNN can be expressed as:

$$h_v^{(l+1)} = \mathrm{AGG}_{r \in \mathcal{R}} \left( \mathrm{AGG}_{u \in \mathcal{N}_r(v)} f_r(h_u^{(l)}, h_v^{(l)}) \right), \tag{2}$$

where $\mathcal{N}_r(v)$ denotes the neighbors of node $v$ under relation type $r$, $f_r(\cdot)$ is a relation-specific message function, and AGG is an aggregation operator such as mean or sum pooling.

## 3. What Makes a Desired Graph?

RDBs are engineered for data integrity and storage efficiency, not for representation learning (Fagin, 1979). Consequently, a heterogeneous graph derived mechanically

from an RDB schema is rarely the optimal topology for downstream prediction. It often suffers from two opposing pathologies: *information overload* and *semantic fragmentation*. In this section, we investigate the fundamental question: *What makes a desired graph for RDL?* We view graph construction as a controlled process and probe structural effects via two operators: "information filtering" and "structural injection".

### 3.1. Information Filtering

Filtering aims to remove task-irrelevant entities and attributes. In RDL, additional types/columns can introduce nuisance variables and increase the effective search space the GNN must traverse. Thus, we treat filtering as a mechanism for structural capacity control: it constrains which parts of the schema-derived representation are even available to the learner. We implement filtering via a simple hierarchical gating design that can suppress redundancy at two granularities.

**Column-Level Filtering.** For a node $v$ with type $\phi(v) = T_i$, we introduce a dedicated learnable gating vector $g_{T_i} \in [0, 1]^{d_i}$, where $d_i$ denotes the dimensionality of the feature vector. This vector is applied element-wise to the initial feature $h_v^{(0)}$:

$$\hat{h}_v = g_{T_i} \odot h_v^{(0)}, \quad \forall v \text{ with } \phi(v) = T_i. \quad (3)$$

To allow near-discrete selection while preserving differentiability, we use a Hard-Concrete relaxation (Louizos et al., 2017):

$$g_{T_i} = \sigma\left(\frac{\log u - \log(1 - u) + \theta_{T_i}}{\tau}\right) \cdot (\zeta - \gamma) + \gamma, \quad (4)$$
$$u \sim \text{Uniform}(0, 1)^{d_i}.$$

Here $\theta_{T_i}$ are learnable logits, $\tau$ controls the temperature. Following (Louizos et al., 2017), the gates are stretched to $(\gamma, \zeta)$ and then clipped to the $[0, 1]$. During inference, deterministic gating is applied as $g_{T_i} = \mathbb{I}[\sigma(\theta_{T_i}) > \tau_g]$, where threshold $\tau_g$ is a hyperparameter.

**Table-Level Filtering.** At a coarser scale, a scalar gating variable $s_{T_i} \in [0, 1]$ is assigned to each node type $T_i$ and applied to the column-filtered output:

$$\tilde{h}_v = s_{T_i} \cdot \hat{h}_v. \quad (5)$$

Similar to column gating, $s_{T_i}$ is parameterized by a scalar logit $\vartheta_{T_i}$ via the Hard-Concrete distribution.

### 3.2. Structural Injection

Filtering alone cannot fix semantic fragmentation: when the schema decomposes an interaction into multi-hop chains, a GNN may need excessive depth to recover a dependency. Structural injection addresses this by adding edges that repair missing relational motifs. However, a key subtlety is

that injection is not a free lunch: adding edges indiscriminately can distort neighborhoods and introduce harmful inductive bias. Therefore, we treat injection strategies as controlled, interpretable probes, and empirically ask when each prior helps. We consider four generic "semantic repair" strategies:

**Type-wise KNN (Homophily Repair).** RDB tables often lack explicit links between similar entities. To restore local neighborhoods, we connect each node to its top-$n$ most similar peers within the same type $T_i$:

$$\mathcal{E}_{knn}^{(i)} = \{(u, v) \mid v \in \text{Top-n}_{v' \in T_i}(\text{sim}(h_u^{(0)}, h_{v'}^{(0)}))\}, \quad (6)$$

where $\text{sim}(\cdot, \cdot)$ denotes cosine similarity. This reinforces intra-type homophily.

**Two-hop Shortcuts (Topological Repair).** To mitigate signal attenuation across long relational chains $A \xrightarrow{r_1} B \xrightarrow{r_2} C$, we introduce direct shortcut edges:

$$\mathcal{E}_{2hop} = \{(u_a, u_c) \mid \exists u_b \text{ s.t. } (u_a, u_b) \in \mathcal{E}_{r_1} \wedge (u_b, u_c) \in \mathcal{E}_{r_2}\}. \quad (7)$$

This explicitly reduces the message-passing depth for high-order dependencies.

**Behavioral Similarity (Collaboration Repair).** For node types $A, B$ connected by relation $r$, we aggregate each $A$-type node's neighbors of type $B$ and compute pairwise similarity:

$$\mathcal{E}_{\text{sim}} =$$
$$\{(v_{a_i}, v_{a_j}) \mid \text{sim}(f_{agg}(\mathcal{N}_B^r(v_{a_i})), f_{agg}(\mathcal{N}_B^r(v_{a_j}))) > \tau_{\text{sim}}\}, \quad (8)$$

where $\mathcal{N}_B^r(v_a)$ denotes the set of $B$-type neighbors of $v_a$ connected via relation $r$, $f_{agg}$ is an aggregation function, and $\tau_{\text{sim}}$ is a similarity threshold.

**Temporal Continuity (Causal Repair).** RDBs store states as discrete rows. To capture causal evolution, we link time-indexed nodes of an entity in their natural temporal order. For a sequence $[v_1, \ldots, v_m]$ sorted by time, we define:

$$E_{\text{time}} = \{(v_{t-1}, v_t) \mid t = 2, \ldots, m\}. \quad (9)$$

This allows the graph to explicitly model state transitions $(v_{t-1} \rightarrow v_t)$. This template is only instantiated for node types with a timestamp column, and edges connect records of the *same entity* across time.

A single strategy can be instantiated at multiple compatible node-type configurations; we refer to each realization as a "template instance" and isolate the effect of one strategy per analysis experiment by enabling only its template set. For a fixed strategy $s$, its instantiation over the graph yields a finite set of templates: $\mathcal{M}(s) = \{\mathcal{E}_k^{(s)}\}_{k=1}^{K_s}$, where each $\mathcal{E}_k^{(s)}$ represents a concrete injected edge set and $K_s = |\mathcal{M}(s)|$ is the number of instantiated templates. The resulting augmented graph is given by: $\mathcal{E}' = \mathcal{E} \cup \bigcup_{k=1}^{K_s} \mathcal{E}_k^{(s)}$.

*Table 1.* Performance of information filtering. We highlight the best result in **bold** and the second best in *italics*.

| Methods | study-outcome(↑) | driver-top3(↑) | user-repeat(↑) | user-clicks(↑) | study-adverse(↓) | driver-position(↓) | user-attendance(↓) | condition-sponsor-run(↑) | post-post-related(↑) |
|---|---|---|---|---|---|---|---|---|---|
| Base | $69.44_{\pm 0.50}$ | $81.48_{\pm 1.06}$ | $78.70_{\pm 0.05}$ | $65.44_{\pm 0.70}$ | $45.50_{\pm 0.14}$ | $3.88_{\pm 0.02}$ | $0.24_{\pm 0.01}$ | $2.83_{\pm 0.01}$ | $1.24_{\pm 0.08}$ |
| RandFilter | $63.21_{\pm 1.14}$ | $75.36_{\pm 1.75}$ | $73.27_{\pm 1.66}$ | $61.54_{\pm 1.82}$ | $48.23_{\pm 1.17}$ | $4.85_{\pm 0.87}$ | $0.47_{\pm 0.05}$ | $2.41_{\pm 0.01}$ | $0.41_{\pm 0.08}$ |
| VarFilter | *$69.91_{\pm 0.43}$* | $82.07_{\pm 0.81}$ | $78.16_{\pm 0.12}$ | $66.04_{\pm 0.57}$ | $45.18_{\pm 0.11}$ | *$3.51_{\pm 0.03}$* | $0.24_{\pm 0.01}$ | $2.91_{\pm 0.01}$ | $1.63_{\pm 0.05}$ |
| ColFilter | $69.87_{\pm 0.01}$ | *$83.64_{\pm 0.04}$* | *$79.95_{\pm 0.82}$* | *$66.31_{\pm 0.18}$* | **$43.83_{\pm 0.38}$** | $3.83_{\pm 0.03}$ | **$0.23_{\pm 0.01}$** | *$3.12_{\pm 0.01}$* | *$1.74_{\pm 0.01}$* |
| FullFilter | **$70.85_{\pm 0.46}$** | **$84.66_{\pm 0.61}$** | **$80.47_{\pm 0.89}$** | **$67.17_{\pm 0.06}$** | *$43.60_{\pm 0.05}$* | $3.79_{\pm 0.16}$ | **$0.23_{\pm 0.01}$** | **$3.63_{\pm 0.01}$** | **$1.77_{\pm 0.01}$** |

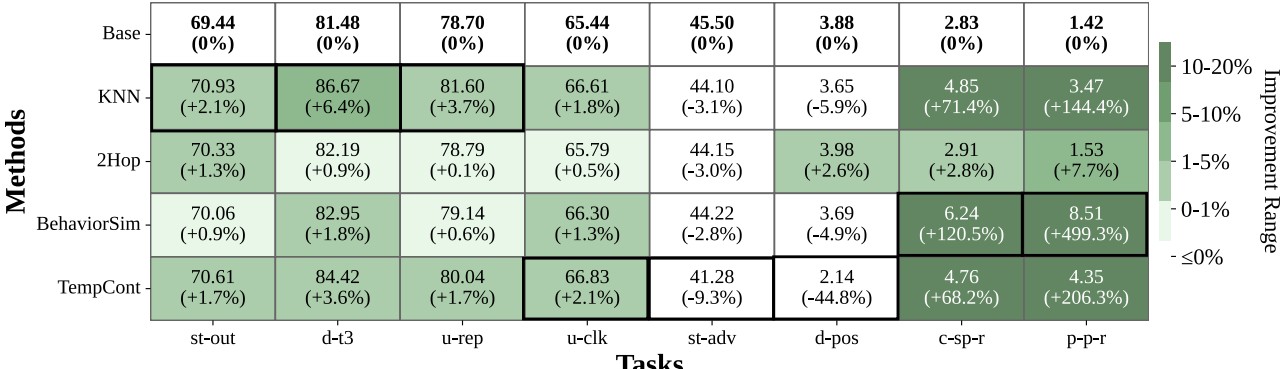

*Figure 1.* Performance gains from structural injection across tasks. Black boxes highlight the best score for each task. Due to space limitations, we use abbreviations for all tasks.

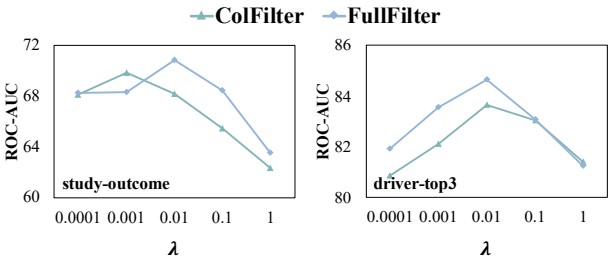

*Figure 2.* Effects of filtering strength on study-outcome and driver-top3 tasks.

### 3.3. Empirical Analysis: Structural Probes for Desired Graphs

We conduct controlled experiments on 9 tasks from REL-Bench (Robinson et al., 2024) to characterize how these structural primitives affect learning. By observing the behavior of these probes, we derive key insights into the anatomy of a desired graph. The statistics and details of these datasets and tasks can be found in Appendix A.

#### 3.3.1. SEMANTIC SPARSIFICATION BEATS STATISTICAL PRUNING

Table 1 contrasts different filtering approaches. A particularly revealing comparison is variance-based filtering (VarFilter) versus semantic gating. VarFilter keeps high-variance features, yet it does not consistently improve over the base graph and can underperform stronger semantic filters. This highlights a core property of relational data: many high-variance attributes are operational artifacts that are statistically "active" but causally irrelevant to the label. In other words, variance is not predictive signal in RDBs; filtering must reflect semantic relevance to the task. Next,

Figure 2 shows a consistent non-monotonic trend as we increase filtering strength (by varying the sparsity weight $\lambda$). Performance first improves as nuisance structure is removed, peaks at an intermediate regime, and then degrades when the filtering begins to erase task-relevant evidence. This inverted-U behavior means the desired graph is not the sparsest graph, nor the fullest schema graph, but the one that strikes a task-dependent balance between denoising and information retention. Overall, when optimizing schema-derived graphs, treat filtering as a bias–variance knob: insufficient filtering leaves semantic noise that can cause oversmoothing and signal dilution, while excessive filtering induces under-specification by removing useful relational evidence.

#### 3.3.2. INJECTION HELPS WHEN IT INTRODUCES THE RIGHT RELATIONAL MOTIF

Figure 1 provides a complementary picture for injection. A naive expectation might be that adding edges increases expressivity and thus helps performance. The data contradicts that: injection is beneficial only for certain task families, and can be neutral or negative when the injected motif does not match what the label depends on. We intentionally proposed multiple, qualitatively different repair motifs and used the results to infer task–motif regularities: (1) Classification tasks tend to benefit from homophily repair (type-wise KNN), consistent with the need to reinforce local cluster structure for separability. (2) Regression tasks show strong gains from temporal continuity, indicating that explicit causal chains provide the most relevant inductive bias. (3) Recommendation tasks benefit most from behavioral similarity, aligning with the need to expose collaborative signals that are oth-

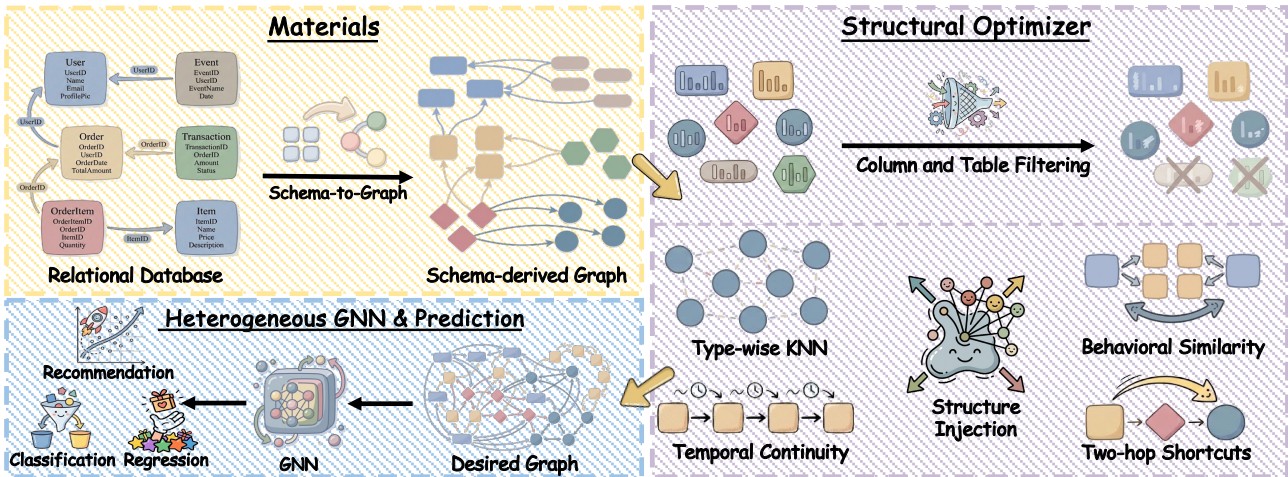

*Figure 3.* Overview of the unified structural optimizer. The framework takes a relational database, converts it into heterogeneous graphs, and applies two complementary structural operations, namely information filtering and structural injection, before passing the optimized graph into a heterogeneous GNN for end-to-end prediction.

erwise fragmented across join paths. Equally important is the negative evidence: applying a mismatched motif can degrade performance (e.g., adding shortcuts to temporal tasks), which implies that the desired graph is not characterized by edge density, but by whether the topology injects task-relevant inductive bias. Overall, structural injection should be treated as motif selection: add edges to expose the dependency pattern the task appears to require.

Filtering and injection are complementary, filtering removes weakly relevant structure, while injection selectively adds relational "shortcuts". A desired graph emerges when these two operations jointly move the schema-derived graph away from overload and fragmentation toward a compact topology whose neighborhoods reflect task-relevant dependencies. More analysis of the effects of filtering and injection strength can be found in Appendix D.

### 3.4. Theoretical Principles

We formalize the notion of a *desired graph* and connect it to the two structural operators studied in Section 3.3. Let $\phi_g$ denote structural parameters (feature/type/template gates) that induce a graph view $G_{\phi_g}$ from a schema-derived graph $G$, and let $\theta$ denote backbone parameters. In particular, we define the regularized risk:

$$\mathcal{R}(\phi_g, \theta) = \mathbb{E}\big[\mathcal{L}_{\text{task}}(Y, \hat{Y}_{\phi_g, \theta})\big] \\ + \beta_{\text{vib}} \mathbb{E}\Big[\text{KL}\big(q(z_v \mid \tilde{h}_v) \,\|\, \mathcal{N}(0, I)\big)\Big], \quad (10)$$

and the structural complexity:

$$\Omega(\phi_g) = \lambda \sum_{T_i \in \mathcal{T}} \mathbb{E}\big[\|g_{T_i}\|_0\big] + \lambda_s \sum_{T_i \in \mathcal{T}} \mathbb{E}\big[\|s_{T_i}\|_0\big] \\ + \lambda_k \sum_{k=1}^{K_{\text{temp}}} \mathbb{E}\big[\|g_k\|_0\big], \quad (11)$$

where $g_{T_i}$ and $s_{T_i}$ are the column-level and table-level gates of table $T_i$, and $g_k$ is the gate associated with the $k$-th injection template. For a fixed structure $\phi_g$, the best achievable population risk is:

$$\mathcal{R}^{\star}(\phi_g) := \inf_{\theta} \mathcal{R}(\phi_g, \theta). \quad (12)$$

**Definition 3.1** (Desired graph). A desired graph for a given task is any graph view $G_{\phi_g^{\star}}$ whose structure parameter $\phi_g^{\star}$ solves:

$$\phi_g^{\star} \in \arg\min_{\phi_g} \Big\{ \mathcal{R}^{\star}(\phi_g) + \Omega(\phi_g) \Big\}. \quad (13)$$

Intuitively, a desired graph is one that (i) supports good task performance after optimizing the encoder parameters $\theta$, and (ii) is structurally simple according to the sparsity-promoting regularizer $\Omega(\phi_g)$.

The regularized risk (Eq. (10)) includes a VIB term $\beta_{\text{vib}}\mathbb{E}[\text{KL}(\cdots)]$, which provides continuous information compression complementary to discrete gate-based filtering. We defer the VIB module design to Section 4.1, where the complete unified optimizer is presented.

**Filtering as structural capacity control.**

Let $M = (M_1, \ldots, M_J)$ denote the collection of Bernoulli gates (covering attribute-, type-, and optionally template-level selection), and let $\mu := \mathbb{E}\|M\|_0$ be the expected number of active components. The following theorem shows that expected sparsity controls an explicit upper bound on the entropy of gating configurations, hence restricting the structural hypothesis space.

**Theorem 3.2** (Filtering controls structural capacity). *Let $M = (M_1, \ldots, M_J)$ be Bernoulli gates with $\pi_j = \mathbb{P}(M_j = 1)$ and $\mu = \sum_{j=1}^{J} \pi_j = \mathbb{E}\|M\|_0$. Then $H(M) \leq \sum_{j=1}^{J} h(\pi_j) \leq J h\big(\frac{\mu}{J}\big)$, where $H(\cdot)$ is Shannon entropy and $h(\cdot)$ is binary entropy. In the sparse regime $\mu/J \leq 1/2$*

*(encouraged by $\ell_0$ penalties), the bound $J\,h(\mu/J)$ is monotone increasing in $\mu$.*

The proof is in Appendix H. This provides a theoretical lens for the inverted-U behavior in Section 3.3: too little filtering preserves a large structural space where nuisance configurations persist, while too much filtering overly restricts structure and removes useful evidence.

**Injection as search-space enrichment.** Structural injection introduces additional candidate edges via instantiated templates. When template gates are learnable (and penalized in $\Omega(\phi_g)$ via Eq. (11)), injection enlarges the feasible family of structures; in an idealized population setting, this cannot worsen the best achievable regularized objective in Eq. (13), since the optimizer can always suppress harmful templates by setting their gates to zero. We formalize this monotonicity in Appendix H.

## 4. A Unified Structural Optimizer

Section 3 characterizes *what makes a desired graph* by treating graph construction as a controlled process with two operators: information filtering and structural injection. Here we provide a simple end-to-end instantiation that operationalizes these operators. Our goal in this section is not to claim a uniquely optimal optimizer design, but to show that the empirical regularities can be exploited automatically by learning (i) *what to remove* and (ii) *what to add* under a unified objective. Our optimizer is deliberately constrained to use interpretable templates rather than free-form edge parameters. This design choice prioritizes scientific interpretability over maximum flexibility: each template has semantic identity (homophily repair, temporal continuity, etc.), allowing us to extract reusable findings about task-motif regularities. An unconstrained structure learner could achieve higher numbers but would produce opaque structures from which no generalizable principles can be extracted. An overview of the architecture is shown in Figure 3.

Given the relational-to-graph transformation in Section 2, the model operates on temporally consistent subgraphs. Initial node embeddings are obtained via the heterogeneous encoder in Eq. (1). These embeddings are then refined by the two operators. The resulting graph is processed by a heterogeneous GNN, followed by a multi-layer perceptron (MLP) (LeCun et al., 2015) head for prediction.

### 4.1. Information Filtering

The information filtering module implements the dual-level sparsification defined in Section 3.1. It operates on the encoded features $h_v^{(0)}$ and applies column- and table-level gates based on the Hard-Concrete relaxation. Concretely, we reuse the gating functions in Eqs. (3)–(5): column-level

gates $g_{T_i}$ suppress noisy feature dimensions, and type-level gates $s_{T_i}$ modulate the contribution of each node type.

To complement discrete sparsification with continuous compression, we apply a type-wise Variational Information Bottleneck (VIB) on the filtered representations. For each node $v$ of type $T_i$, the VIB module maps $\tilde{h}_v$ to a stochastic latent representation $z_v$:

$$z_v = \boldsymbol{\mu}(\tilde{h}_v) + \boldsymbol{\sigma}(\tilde{h}_v) \odot \boldsymbol{\epsilon}, \qquad \boldsymbol{\epsilon} \sim \mathcal{N}(\mathbf{0}, \mathbf{I}), \qquad (14)$$

where $\boldsymbol{\mu}(\cdot)$ and $\boldsymbol{\sigma}(\cdot)$ are learnable functions producing the mean and standard deviation of the latent distribution. During inference, we use the mean $z_v = \boldsymbol{\mu}(\tilde{h}_v)$.

This VIB layer provides a continuous information bottleneck on top of the hard-concrete gates, encouraging compact, task-relevant representations while leaving structural decisions to the gating module.

### 4.2. Structural Injection

The structural injection module enhances the relational graph by adding auxiliary edges that improve information propagation. It complements feature-level filtering by addressing connectivity limitations arising from schema-based graphs. We implement the four strategies introduced in Section 3.2: type-wise KNN, two-hop shortcuts, behavioral similarity, and temporal continuity. Each strategy $s \in \mathcal{S}$ yields a set of instantiated templates:

$$\mathcal{M}(s) = \{\mathcal{E}_k^{(s)}\}_{k=1}^{K_s}, \qquad (15)$$

and the full candidate set is:

$$\mathcal{M}_{\text{all}} = \bigcup_{s \in \mathcal{S}} \mathcal{M}(s). \qquad (16)$$

To adaptively regulate the contribution of each template, we associate every $\mathcal{E}_k \in \mathcal{M}_{\text{all}}$ with a learnable gate $g_k \in [0, 1]$. We parameterize each template gate $g_k$ using the same Hard-Concrete relaxation as in Eqs. (3)-(5), so that $g_k$ acts as a stochastic gate during training and its expected activation $\mathbb{E}[\|g_k\|_0]$ admits an $\ell_0$-style sparsity penalty. Let $g = (g_1, \ldots, g_K)$ denote all structural gates. During training, we retain differentiability by using $g_k$ as a continuous weight on template-specific messages rather than performing hard selection. For template $\mathcal{E}_k$, the aggregated message at layer $l$ is:

$$z_{v,k}^{(l)} = g_k \cdot \text{AGG}\big(z_u^{(l)} : u \in \mathcal{N}_k(v)\big), \qquad (17)$$

where $\mathcal{N}_k(v)$ is the neighbor set induced by $\mathcal{E}_k$ and AGG is an aggregation operator (e.g., mean or sum). At inference time, we switch to deterministic selection for efficiency:

$$\mathcal{E}' = \mathcal{E} \cup \{\mathcal{E}_k \mid g_k > \tau_k\}, \qquad (18)$$

where $\tau_k$ is a threshold hyperparameter, and retaining only templates with $g_k > \tau_k$ results in a sparse augmented graph.

## 4.3. Integrated Graph Representation Learning

The filtered features and injected structure are combined in the final representation learning phase. The latent representations $z_v$ serve as node features, and the edge set $\mathcal{E}'$ defines the connectivity. A GNN then performs message passing over both original and template-induced relations.

We treat each instantiated structural template as defining a distinct relation type. Let $\mathcal{R}$ denote the set of original relation types and $\mathcal{R}_{\text{template}}$ includes all instantiated templates $\mathcal{E}_k$. For node $v$ at layer $l$, the update rule aggregates over all relation types:

$$z_v^{(l+1)} = \text{AGG}_{r \in \mathcal{R} \cup \mathcal{R}_{\text{template}}} \left( \text{AGG}_{u \in \mathcal{N}_r(v)} f_r(z_u^{(l)}, z_v^{(l)}) \right), \tag{19}$$

where $\mathcal{N}_r(v)$ denotes the neighbors of $v$ under relation $r$. For template-induced relations, message aggregation is modulated by the corresponding structural gate $g_k$ as defined in Eq. (17). The final node representations $z_v^{(L)}$ are fed to an MLP head:

$$\hat{y}_v = \text{MLP}(z_v^{(L)}), \tag{20}$$

where $L$ is the maximum depth of the GNN. Finally, the model is trained end-to-end using the following objective:

$$\begin{aligned}
\mathcal{L} = {}& \mathcal{L}_{\text{task}}(\hat{Y}, Y) \\
& + \beta_{\text{vib}} \sum_{T_i \in \mathcal{T}} \frac{1}{|T_i|} \sum_{v \in T_i} \text{KL}\left( q(z_v \mid \tilde{h}_v) \,\|\, \mathcal{N}(\mathbf{0}, \mathbf{I}) \right) \\
& + \lambda \left( \sum_{T_i \in \mathcal{T}} \mathbb{E}[\|g_{T_i}\|_0] + \lambda_s \sum_{T_i \in \mathcal{T}} \mathbb{E}[\|s_{T_i}\|_0] \right) \\
& + \lambda_k \sum_{k=1}^{K_{\text{temp}}} \mathbb{E}[\|g_k\|_0],
\end{aligned} \tag{21}$$

where $\mathcal{L}_{\text{task}}$ is the supervised loss, $\beta_{\text{vib}}$, $\lambda$, $\lambda_s$, and $\lambda_k$ are hyperparameters, and $K_{\text{temp}}$ denotes the total number of instantiated structural templates.

## 5. Related Work

### 5.1. Traditional Methods on Relational Databases

The conventional approach to predictive modeling on RDBs involves two distinct steps: feature engineering via SQL-based data integration from multiple tables, followed by training standard tabular models (Ke et al., 2017; Chen, 2016). XGBoost (Chen, 2016) is an advanced gradient boosting framework that enhances model performance by sequentially building decision trees. It is renowned for achieving high predictive accuracy while maintaining exceptional computational speed and scalability, making it a benchmark algorithm for tabular data. LightGBM (Ke et al., 2017) is a gradient boosting framework designed for high performance and scalability. By leveraging histogram-based

algorithms and innovative sampling techniques, it delivers exceptional training speed and efficiency. A key limitation of this two-step paradigm is its inability to capture the rich structural information that inherently connects database entities.

### 5.2. Deep Learning on Relational Databases

RDL aims to directly apply deep neural networks to the structured data in relational databases (Cvitkovic, 2020; Fey et al., 2024; Chen et al., 2025; Šír, 2021; Zahradník et al., 2023; Kanatsoulis et al., 2025). Its core idea is to learn complex patterns and dependencies from multiple interrelated tables in an end-to-end manner. Fey et al. (Fey et al., 2024) introduces a blueprint for end-to-end learning on relational databases. It represents the relational database as a temporal heterogeneous graph, and then a graph neural network learns to leverage representations from all input data. REL-GNN (Chen et al., 2025) introduces atomic paths to enable direct single-hop interactions between source and target nodes, and designs a new composite message-passing and graph attention mechanism, thus reducing redundancy and enhancing prediction accuracy. RelGT (Dwivedi et al., 2025) is the first graph transformer architecture specifically designed for relational tables. It employs a novel multi-element tokenization strategy capable of efficiently encoding heterogeneity, temporality, and topology without requiring expensive pre-computation. However, the performance of these RDL models is fundamentally bounded by the initial relational-to-graph conversion. Prevailing mechanical mapping strategies often produce graphs with high complexity and redundancy, which in turn cripples downstream learning from the very beginning.

### 5.3. Structural Optimization in Heterogeneous Graphs

Existing methods for structural optimization in heterogeneous graphs (Zhao et al., 2021; Yun et al., 2019; Schlichtkrull et al., 2018; Wang et al., 2019; You et al., 2021) generally aim to enhance graph quality through topology refinement, often by adjusting edge weights or learning latent structures. For instance, GTN (Yun et al., 2019) introduces an approach to automatically generate useful meta-paths and identify soft connections between nodes, effectively learning a new graph topology that is more conducive to message passing. Meanwhile, HGSL (Zhao et al., 2021) jointly learns both the graph structure and node representations in an end-to-end manner, adapting the connectivity pattern by injecting new edges and refining existing ones based on task-specific objectives. HAN (Wang et al., 2019) uses meta-path–guided attention to enhance semantic connectivity, serving as a classical heterogeneous structure optimizer that strengthens predefined relational semantics. A key limitation of these approaches is their underlying assumption that the input graph possesses a semantically

*Table 2.* Overall predictive performance (ROC-AUC (%), ↑) on classification tasks. We highlight the best result in **bold** and the second best in *italics*.

| Method | study-outcome | driver-dnf | driver-top3 | user-repeat | user-ignore | user-clicks | user-visits |
|---|---|---|---|---|---|---|---|
| Base | $69.44_{\pm0.53}$ | $74.53_{\pm0.27}$ | $81.48_{\pm1.06}$ | $78.75_{\pm0.62}$ | $75.39_{\pm0.54}$ | $65.44_{\pm0.71}$ | $63.82_{\pm0.67}$ |
| LightGBM | $65.27_{\pm0.81}$ | $69.13_{\pm1.14}$ | $76.09_{\pm0.93}$ | $74.51_{\pm0.72}$ | $74.58_{\pm0.71}$ | $61.27_{\pm0.93}$ | $63.03_{\pm0.85}$ |
| HAN | $69.05_{\pm0.67}$ | $74.05_{\pm1.08}$ | $82.09_{\pm0.93}$ | $79.24_{\pm0.75}$ | $79.27_{\pm0.76}$ | $66.15_{\pm0.63}$ | $66.57_{\pm0.74}$ |
| GTN | $68.15_{\pm0.47}$ | $72.36_{\pm0.68}$ | $82.50_{\pm0.63}$ | $79.85_{\pm0.61}$ | $78.62_{\pm0.77}$ | $64.85_{\pm0.69}$ | $66.82_{\pm0.34}$ |
| HGSL | $69.51_{\pm0.47}$ | $71.56_{\pm0.48}$ | $83.51_{\pm0.75}$ | $80.64_{\pm0.81}$ | $77.23_{\pm1.07}$ | $63.42_{\pm0.86}$ | $67.08_{\pm0.71}$ |
| REL-GNN | $70.86_{\pm0.54}$ | $75.41_{\pm0.29}$ | $82.71_{\pm0.83}$ | $79.08_{\pm0.81}$ | $80.31_{\pm0.86}$ | $65.91_{\pm0.82}$ | $68.14_{\pm0.76}$ |
| RelGT | $69.74_{\pm0.32}$ | $76.33_{\pm0.58}$ | $83.14_{\pm0.75}$ | $79.46_{\pm0.32}$ | $81.24_{\pm0.77}$ | $66.87_{\pm0.60}$ | $68.94_{\pm0.65}$ |
| LLM-Struct | $70.83_{\pm0.48}$ | $76.80_{\pm0.39}$ | $83.57_{\pm0.37}$ | $80.13_{\pm0.77}$ | $79.90_{\pm0.73}$ | $\mathbf{68.75_{\pm0.37}}$ | $69.25_{\pm0.39}$ |
| Ours | $\mathbf{72.35_{\pm0.37}}$ | $\mathbf{78.21_{\pm0.15}}$ | $\mathbf{86.82_{\pm0.61}}$ | $\mathbf{82.36_{\pm0.55}}$ | $\mathbf{81.25_{\pm0.64}}$ | $67.53_{\pm0.15}$ | $\mathbf{70.15_{\pm0.56}}$ |

*Table 3.* Overall predictive performance (MAE, ↓) on regression tasks. Best results are in **bold**, second best are in *italics*.

| Method | study-adverse | site-success | driver-position | user-attendance | ad-ctr |
|---|---|---|---|---|---|
| Base | 45.50 | 38.86 | 3.88 | 0.24 | 0.231 |
| LightGBM | 48.31 | 0.38 | 4.95 | 0.51 | 0.242 |
| HAN | 45.16 | 0.35 | 3.76 | 0.24 | 0.228 |
| GTN | 44.82 | 0.36 | 3.70 | 0.23 | 0.225 |
| HGSL | 44.69 | 0.36 | 3.85 | 0.25 | 0.246 |
| REL-GNN | 44.93 | 0.34 | 3.78 | 0.23 | 0.227 |
| RelGT | 44.62 | 0.32 | 3.72 | 0.23 | 0.225 |
| LLM-Struct | 44.73 | 0.34 | 3.69 | 0.23 | 0.228 |
| Ours | **41.28** | 0.31 | **2.13** | 0.23 | **0.221** |

*Table 4.* Overall predictive performance on recommendation tasks (MAP, ↑). Best results are in **bold**, second best in *italics*.

| Method | user-ad-visit | user-post-comment | post-post-related | condition-sponsor-run | site-sponsor-run |
|---|---|---|---|---|---|
| Base | 2.31 | 8.77 | 1.24 | 2.83 | 10.21 |
| LightGBM | 1.14 | 6.26 | 0.41 | 2.15 | 8.35 |
| HAN | 2.33 | 11.06 | 1.31 | 2.91 | 15.27 |
| ID-GNN | 3.24 | 13.41 | 10.61 | 11.65 | 18.25 |
| REL-GNN | 2.87 | 13.12 | 2.43 | 3.47 | 17.31 |
| LLM-Struct | 3.28 | 13.64 | 4.72 | 4.62 | 18.71 |
| Ours | **3.84** | **14.82** | 8.53 | **6.24** | **19.15** |

meaningful base topology to be refined. However, in graphs derived from relational databases, connectivity is dictated by schema design and foreign-key constraints rather than task relevance. This results in node and relation types that reflect data organization principles instead of task semantics.

# 6. Experiments

## 6.1. Experimental Setup

**Datasets and tasks.** We use the same RELBench tasks and evaluation protocol as in Section 3.3, covering entity classification, regression, and recommendation. Dataset statistics are deferred to Appendix A.

**Evaluation.** We evaluate predictive performance using task-specific metrics. For entity classification, we report the Area Under the ROC Curve (ROC-AUC) (Hanley & McNeil, 1983), where higher values indicate better perfor-

mance. For entity regression, we use Mean Absolute Error (MAE), where lower values indicate better performance. For recommendation tasks, we report Mean Average Precision (MAP), where higher values correspond to better ranking quality. All reported results are averaged over five independent runs with different random seeds.

**Baselines.** We compare our unified structural optimizer with five groups of baselines. **(1) Base relational GNN pipeline:** Base. **(2) Traditional non-graph methods:** LightGBM (Ke et al., 2017). **(3) Heterogeneous graph structure optimization methods:** HAN (Wang et al., 2019), ID-GNN (You et al., 2021), GTN (Yun et al., 2019) and HGSL (Zhao et al., 2021). **(4) RDL methods:** REL-GNN (Chen et al., 2025) and Relational Graph Transformer (RelGT) (Dwivedi et al., 2025). **(5) LLM-based structural optimization:** LLM-Struct (Achiam et al., 2023). All methods use identical temporal splits, feature encoders, and GNN backbones where applicable. For HAN, we construct simple meta-paths directly from the database schema.

We defer details on datasets (A), baselines (B), experimental setup (C), effects of filtering strength and injection strength (D), overall predictive performance on large-scale datasets (E), ablation study (F) and choice of structural complexity metri (G) to the Appendix.

## 6.2. Overall Predictive Performance

Tables 2, 3, and 4 report results on classification, regression, and recommendation tasks. Three consistent patterns emerge. (1) Our method achieves the best or near-best performance across almost all tasks. This is because, as predicted by our analysis in Section 3.3, filtering removes task-irrelevant structure that introduces noise and over-smoothing, while structural injection exposes task-relevant dependencies that are otherwise fragmented by the schema. As a result, our graphs provide cleaner signals for classification, lower-variance features for regression, and more coherent interaction structure for recommendation. (2) Traditional RDL and heterogeneous GNN baselines (e.g., HAN,

REL-GNN, RelGT) yield only moderate improvements over the Base model. Although these methods improve message passing, they operate on the same schema-derived graphs and therefore inherit their information overload and fragmented connectivity. When these structural bottlenecks dominate, architectural improvements alone cannot recover the missing or diluted task-relevant information. (3) LLM-based structural optimization is competitive but inconsistent. While LLMs can propose reasonable schema-level modifications, they cannot exploit fine-grained instance-level patterns such as interaction frequency or temporal continuity, leading to injections that are often misaligned with the learning objective. In contrast, our optimizer is guided directly by task-aware gradients, allowing it to identify structural changes that consistently improve predictive performance. Overall, these results confirm that principled structural optimization is effective across task types.

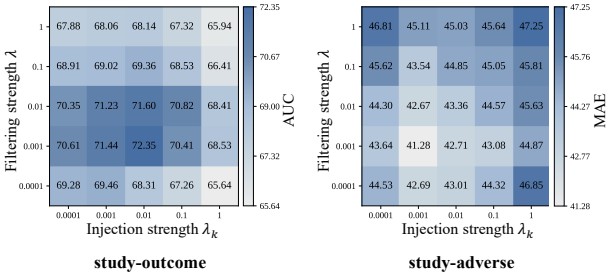

**Figure 4.** Trade-off between information filtering and structural injection on study-outcome and study-adverse tasks.

## 6.3. Trade-off Between Filtering and Injection

To study how filtering and structural injection interact, we vary two regularization coefficients: $\lambda$, which controls filtering strength, and $\lambda_k$, which controls the sparsity of injected templates. Figure 4 shows the resulting performance landscape on the study-outcome and study-adverse tasks. Across both tasks, performance depends non-monotonically on these two factors. When injection is strong but filtering is weak, performance degrades, indicating that excessive added structure amplifies noise. When filtering is strong but injection is weak, performance is also limited, as aggressive pruning removes useful relational signals. The best results appear in a balanced regime where both mechanisms are active. For study-outcome, the highest AUC (72.35) occurs around $\lambda \approx 10^{-3}$ and $\lambda_k \approx 10^{-2}$, while for study-adverse the optimum (41.28) occurs near $\lambda \approx 10^{-3}$ and $\lambda_k \approx 10^{-3}$. In both cases, the optimal region forms a smooth plateau, indicating robustness to small hyperparameter changes.

## 6.4. Inference efficiency

Table 5 reports the average inference time on six downstream tasks. FullFilter consistently achieves the lowest latency due to aggressive structural pruning. While not

**Table 5.** Inference time comparison on six tasks. We report average inference time in seconds (s). Best results are in bold.

| Method | study-outcome | user-engagement | study-adverse | post-votes | condition-sponsor-run | user-item-purchase |
|---|---|---|---|---|---|---|
| Base | 0.55 | 29.01 | 1.06 | 26.90 | 13.71 | 58.86 |
| REL-GNN | 0.64 | 27.93 | 1.75 | 28.06 | 17.24 | 60.48 |
| HAN | 1.15 | 32.56 | 2.90 | 33.65 | 19.04 | 70.85 |
| ColFilter | 0.18 | 14.07 | 0.47 | 13.64 | 5.87 | 15.76 |
| FullFilter | **0.12** | **15.10** | **0.39** | **9.41** | **5.35** | **13.24** |
| Ours | 0.46 | 20.60 | 0.86 | 15.57 | 10.07 | 25.81 |

**Table 6.** Effect of backbone GNNs on predictive performance. Best results are in bold. OOM denotes the out-of-memory error.

| Method | study-outcome ($\uparrow$) | study-adverse ($\downarrow$) | condition-sponsor-run ($\uparrow$) |
|---|---|---|---|
| Base (GraphSAGE) | 69.44 | 45.50 | 2.83 |
| Ours (GraphSAGE) | 72.35 | **41.28** | **6.24** |
| Base (REL-GNN) | 70.86 | 44.93 | 3.47 |
| Ours (REL-GNN) | 71.68 | 42.37 | 4.84 |
| Base (HGT) | 70.18 | 44.60 | OOM |
| Ours (HGT) | **72.84** | 42.55 | OOM |

the fastest, our method maintains competitive and stable inference efficiency, consistently outperforming heavier relational baselines such as HAN and REL-GNN. Compared with filtering-only approaches, our method incurs moderate additional cost from relational modeling, but this overhead remains well controlled. Overall, the results show that our approach strikes a practical balance between predictive expressiveness and inference efficiency.

## 6.5. Robustness to Backbone GNNs

To evaluate robustness across GNN backbones, we replace GraphSAGE with REL-GNN and HGT and test each backbone with and without our structural optimizer on three tasks. As shown in Table 6, the optimizer consistently improves performance across backbones and task types. Gains are largest for GraphSAGE, while REL-GNN shows smaller but still consistent improvements, indicating that relation-aware message passing only partially mitigates structural deficiencies. HGT also benefits from our optimizer on tasks where training is feasible. Overall, these results confirm that structural optimization is backbone-agnostic and complementary to different heterogeneous GNN architectures.

## 7. Conclusion

This work demonstrates that strict adherence to raw database schemas limits predictive performance in RDL. We identify a fundamental structural mismatch between RDB storage efficiency and GNN reasoning requirements, resulting in information overload and semantic fragmentation. By framing graph construction as a structural information bottleneck problem, we show that an optimal graph requires balancing variational compression to eliminate noise with inductive augmentation to restore connectivity. Our unified structural optimizer instantiates these principles, consistently outperforming state-of-the-art baselines across 26 diverse tasks.

## Acknowledgements

This research / project is supported by the National Research Foundation, Singapore, under its Frontier Competitive Research Programme (NRF-F-CRP-2024-0005). Any opinions, findings and conclusions or recommendations expressed in this material are those of the author(s) and do not reflect the views of National Research Foundation, Singapore.

## Impact Statement

This research significantly advances the integration of deep learning with modern data infrastructure by addressing the structural mismatch between relational databases and graph neural networks. By introducing a principled framework to optimize relational graphs through information filtering and task-aligned structural injection, this work enhances the accessibility and efficiency of machine learning for large-scale relational datasets. The proposed structural optimizer not only improves predictive accuracy across diverse domains but also reduces the computational cost of inference by removing task-irrelevant data components. Furthermore, by automating the transformation of raw database schemas into task-dependent topologies, this work reduces the reliance on manual, domain-specific feature engineering, thereby democratizing the application of advanced graph-based reasoning to a wide range of industrial and scientific challenges.

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

*Table 7.* Datasets statistics.

| Dataset | Task | Abbr | Task type | #Rows of training table | | |
| --- | --- | --- | --- | --- | --- | --- |
| | | | | Train | Validation | Test |
| rel-amazon | item-churn | i-ch | classification | 2,559,264 | 177,689 | 166,842 |
| | item-ltv | i-ltv | regression | 2,707,679 | 166,978 | 178,334 |
| | user-item-review | u-i-v | recommendation | 2,324,177 | 116,970 | 127,021 |
| rel-avito | user-clicks | u-clk | classification | 59,454 | 21,183 | 47,996 |
| | user-visits | u-vis | classification | 86,619 | 29,979 | 36,129 |
| | ad-ctr | a-ctr | regression | 5,100 | 1,766 | 1,816 |
| | user-ad-visit | u-a-v | recommendation | 86,616 | 29,979 | 36,129 |
| rel-event | user-repeat | u-rep | classification | 3,842 | 268 | 246 |
| | user-ignore | u-ign | classification | 19,239 | 4,185 | 4,010 |
| | user-attendance | u-att | regression | 19,261 | 2,014 | 2,006 |
| rel-f1 | driver-dnf | d-dnf | classification | 11,411 | 566 | 702 |
| | driver-top3 | d-t3 | classification | 1,353 | 588 | 726 |
| | driver-position | d-pos | regression | 7,453 | 499 | 760 |
| rel-hm | user-churn | u-ch | classification | 3,871,410 | 76,556 | 74,575 |
| | item-sales | i-sal | regression | 5,488,184 | 105,542 | 105,542 |
| | user-item-purchase | u-i-p | recommendation | 3,878,451 | 74,575 | 67,144 |
| rel-stack | user-engagement | u-eng | classification | 1,360,850 | 85,838 | 88,137 |
| | user-badge | u-bdg | classification | 3,386,276 | 247,398 | 255,360 |
| | post-votes | p-vot | regression | 2,453,921 | 156,216 | 160,903 |
| | user-post-comment | u-p-c | recommendation | 21,239 | 825 | 758 |
| | post-post-related | p-p-r | recommendation | 5,855 | 226 | 258 |
| rel-trial | study-outcome | st-out | classification | 11,994 | 960 | 825 |
| | study-adverse | st-adv | regression | 43,335 | 3,596 | 3,098 |
| | site-success | si-suc | regression | 151,407 | 19,740 | 22,617 |
| | condition-sponsor-run | c-sp-r | recommendation | 36,934 | 2,081 | 2,057 |
| | site-sponsor-run | si-sp-r | recommendation | 669,310 | 37,003 | 27,428 |

## A. Datasets

We evaluate our method on **RELBench** (Robinson et al., 2024), a comprehensive benchmark for relational learning derived from real-world, temporal relational databases. It provides a diverse collection of databases and realistic prediction tasks across seven distinct domains: e-commerce, online marketplaces, social event planning, sports analytics, retail, Q&A platforms, and clinical trials. Each dataset features a multi-table schema with temporal foreign-key relationships, enabling the construction of temporal heterogeneous graphs for end-to-end predictive modeling. The benchmark encompasses 26 prediction tasks spanning three fundamental types: entity classification, entity regression, and recommendation.

Table 7 provides detailed statistics for each dataset and task. Below, we briefly describe the domain and core content of each constituent dataset within RELBench.

**rel-amazon.** This dataset models interactions on the Amazon e-commerce platform. It encompasses product metadata (e.g., price, category), user reviews (including rating and text), and user engagement history, supporting tasks related to product recommendation and user behavior analysis.

**rel-avito.** Derived from Avito, a major online classifieds marketplace, this dataset contains user search queries, advertisement characteristics, and contextual metadata for transactions across various categories such as real estate and vehicles. It facilitates tasks like click-through-rate prediction and user activity forecasting.

**rel-event.** Sourced from Hangtime, a mobile app for tracking social plans, this dataset includes user interactions, event metadata, user demographic profiles, and social network connections. It is designed to study how social relationships influence event participation and user engagement.

**rel-f1.** This dataset comprises historical Formula 1 racing data from 1950 onward. It covers drivers, constructors, circuits, and includes granular records of race results, qualifying sessions, practice laps, and pit stops, enabling predictions about race outcomes and driver performance.

**rel-hm.** Capturing customer transactions and product information from H&M's e-commerce platform, this dataset includes customer demographic attributes, detailed product descriptions, and purchase histories, supporting customer- and product-centric prediction tasks.

**rel-stack.** Based on the Stack Exchange network of Q&A websites, this dataset contains detailed activity logs such as user biographies, posts, comments, edits, votes, and question-answer links. It enables the study of user engagement dynamics and content popularity.

**rel-trial.** Aggregated from the Aggregate Analysis of ClinicalTrials.gov (AACT) database, this dataset includes detailed records of clinical trial protocols, study designs, participant demographics, interventions, and outcome measures, serving as a foundation for predicting trial results and safety profiles.

## B. Baselines

We compare our unified structural optimizer with five groups of baselines.

**(1) Base relational GNN pipeline.** This group corresponds to the standard schema-to-graph pipeline without explicit structural optimization:

- **Base.** The relational GraphSAGE (Hamilton et al., 2017) model used in Section 3.3, trained on the original heterogeneous graph obtained from the RDB schema without filtering or structural injection.

**(2) Traditional non-graph methods.** To measure the benefit of explicit relational modeling, we include a strong tabular baseline:

- **LightGBM (Ke et al., 2017).** A gradient boosting decision tree model trained on flattened relational features, ignoring the graph structure.

**(3) Heterogeneous graph structure optimization methods.** This category contains classical heterogeneous GNNs that enhance or reweight connectivity based on schema semantics:

- **HAN** (Wang et al., 2019). A meta-path–based attention network. We construct simple meta-paths directly from the database schema and let HAN learn attention weights over these paths, providing a lightweight handcrafted structural optimization baseline.

- **GTN** (Yun et al., 2019). A graph transformer that automatically generates useful meta-paths by learning to compose adjacency matrices. We initialize GTN with schema-derived relation matrices and allow it to learn multi-hop compositions, providing a learnable alternative to manually specified meta-paths.

- **HGSL** (Zhao et al., 2021). A heterogeneous graph structure learning method that jointly optimizes graph topology and node representations. We apply HGSL to refine the schema-derived graph by learning edge weights and injecting new connections based on feature similarity, serving as a feature-driven structural optimization baseline.

- **ID-GNN** (You et al., 2021) incorporates learnable node identity embeddings, particularly effective for recommendation tasks. We evaluate whether our structural optimizer provides orthogonal gains when applied to ID-GNN.

**(4) Relational deep learning (RDL) methods.** These models are specifically designed for relational databases but do not explicitly optimize the graph structure:

- **REL-GNN** (Chen et al., 2025). A relational GNN that aggregates information along foreign-key relations with task-specific encoders.

- **Relational Graph Transformer (RelGT)** (Dwivedi et al., 2025). A transformer-style relational encoder that applies multi-relation attention over the schema-derived graph.

**(5) LLM-based structural optimization.** Finally, we consider a baseline that uses an external large language model to design the graph structure ahead of training:

- **LLM-Struct (Achiam et al., 2023).** A single LLM module that jointly (i) filters irrelevant tables and columns and (ii) proposes additional task-relevant relations based on the textual schema and task description. The resulting filtered and augmented graph is then trained with the same Base backbone.

*Table 8.* Hyperparameter search space for the unified structural optimizer.

| Hyperparameter | Search Range / Setting |
|---|---|
| Learning rate | $\{$1e-4, 1e-3, 1e-2$\}$ |
| Hidden dimension | $\{$64, 128, 256$\}$ |
| Column-level sparsity $\lambda$ | [1e-4, 1e-2] |
| Table-level sparsity $\lambda_s$ | [1e-4, 1e-2] |
| Structural-template sparsity $\lambda_k$ | [1e-4, 1e-2] |
| VIB coefficient $\beta_{\text{vib}}$ | [1e-4, 1e-2] |
| KNN neighbors $n$ | $\{$1, 5, 10, 15, 20$\}$ |
| Temporal chain length $m$ | $\{$1, 3, 5, 7, 9$\}$ |

## C. Experimental Setup

We implemented all models in PyTorch (Hu et al., 2024) and conducted experiments on RELBench tasks using a single NVIDIA A30 GPU. All GNN-based methods, including our optimizer and RDL baselines, are trained with the Adam optimizer (Kingma, 2014). To ensure fair comparison, every baseline is trained under the same temporal train/validation/test split and uses the same heterogeneous encoder described in Section 2. For methods requiring schema-derived structures (e.g., HAN), we generate meta-paths directly from the RDB schema without task-specific tuning. For the LLM-Struct baseline, we conduct experiments with GPT-4 (Achiam et al., 2023), and the LLM is queried once per task to produce a filtered and augmented graph, which is then used to train the same Base backbone as in Section 3.3. Figure 5 shows the complete prompt template.

Hyperparameters for all GNN-based models are selected through grid search on the validation set, following the same search ranges across methods. For our optimizer, Table 8 summarizes the hyperparameter space, covering gating sparsity coefficients, VIB regularization strength, and GNN depth/hidden dimensions. All results are averaged over five independent runs with different random seeds, and mean performance along with standard deviations are reported.

---

**System Role:** You are an expert in relational database schema design and graph neural network architectures. Your task is to analyze database schemas for machine learning tasks and recommend structural optimizations.

**Your Task:** Given a relational database schema and a prediction task, you need to:
(1) **Filter**: Identify tables and columns that are likely irrelevant to the prediction task.
(2) **Augment**: Propose additional edge types that would improve information flow for graph neural networks.

**Input:**
Prediction Task: {task_description}
Database Schema: {dataset_schema}

**Output (JSON):** {
"keep": {
"table_name_1": ["column1", "column2", ...],
"table_name_2": ["column1", "column2", ...],
...
},
"add_edges": [
{"from": "table_A", "to": "table_B",
"reason": "brief explanation"},
...
]
}

---

*Figure 5.* LLM-Struct prompt template.

## D. Empirical Analysis

### D.1. Effects of Filtering Strength.

Figure 6 examines how filtering strength $\lambda$ affects predictive performance across eight diverse tasks, where $\lambda$ represents the sparsity regularization coefficient in the gating module. The results yield two primary observations: (1) Performance

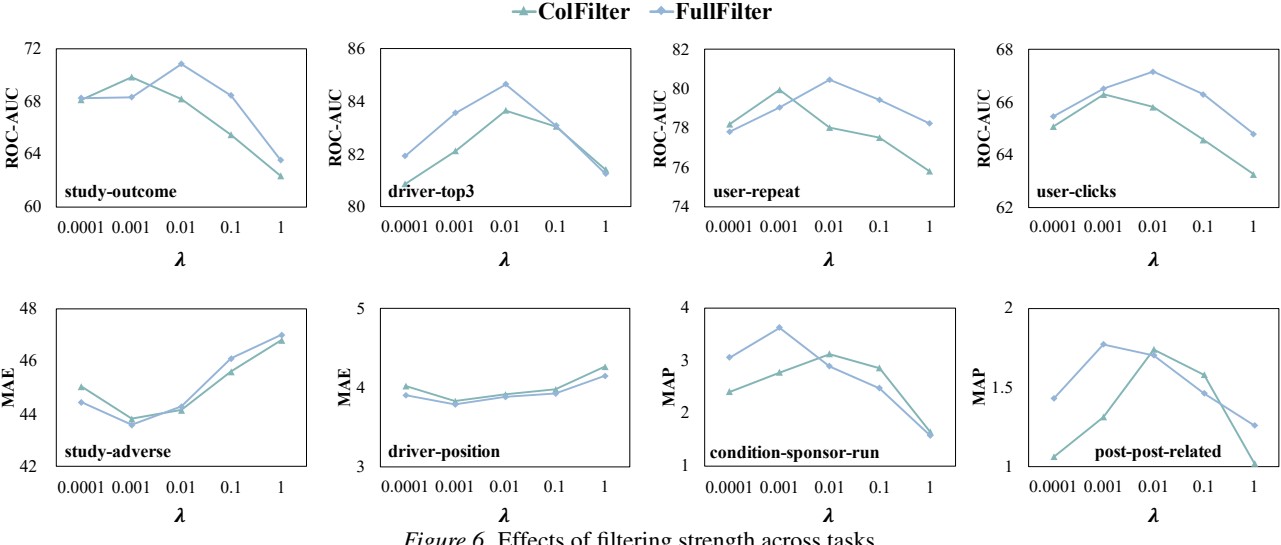

*Figure 6.* Effects of filtering strength across tasks.

follows a consistent non-monotonic pattern. For classification (ROC-AUC) and recommendation (MAP) tasks, we observe an *inverse-U trajectory*: performance improves as $\lambda$ increases from negligible values, peaks at an optimal $\lambda^*$, and then deteriorates as $\lambda$ becomes too aggressive. Conversely, regression tasks (MAE) exhibit a corresponding *U-shaped pattern*, where the error decreases to a minimum before rising again. This confirms the existence of an optimal compression point for all tasks, that effectively suppresses redundancy without discarding task-critical information. (2) The optimal filtering strength $\lambda^*$ is highly task-dependent. The "sweet spot" varies significantly depending on the task semantics. For instance, most classification tasks, such as `driver-top3`, `user-repeat`, and `user-clicks`, align with `study-outcome` in achieving peak performance at a moderate strength ($\lambda \approx 0.01$). In contrast, regression and recommendation tasks like `driver-position` and `condition-sponsor-run` tend to favor a milder filtering strength ($\lambda \approx 0.001$). This sensitivity analysis underscores that filtering is not a "one-size-fits-all" operation; rather, the optimal balance between noise reduction and evidence retention is intrinsically dictated by the specific nature of the downstream prediction task.

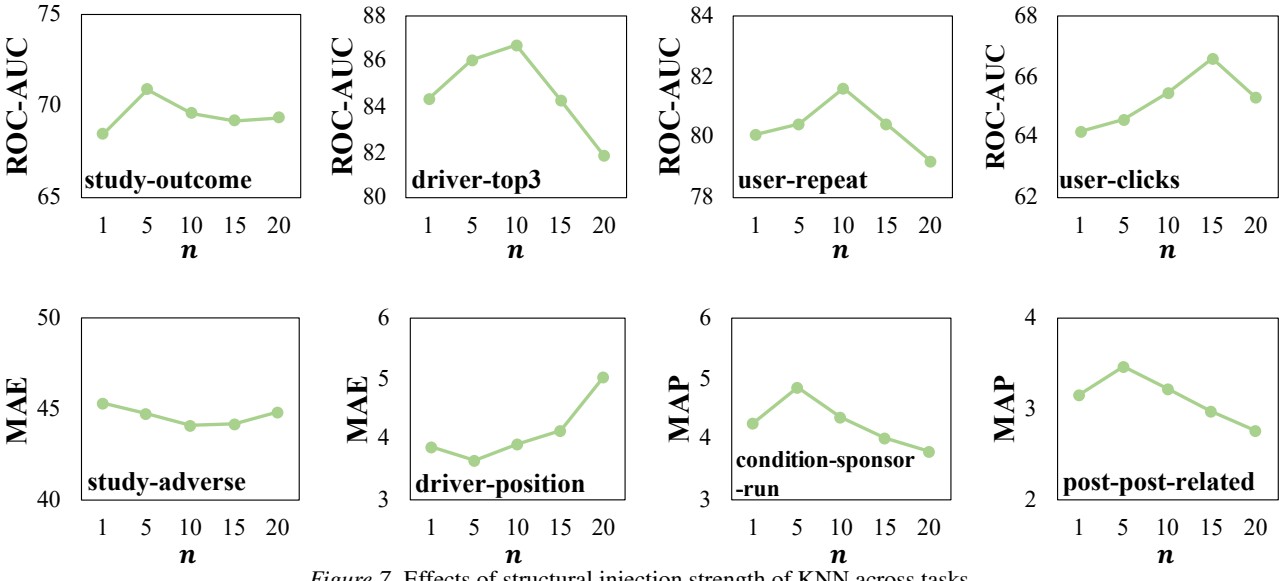

*Figure 7.* Effects of structural injection strength of KNN across tasks.

## D.2. Effects of Injection Strength.

Figure 7 and Figure 8 report how the injection strength affects the performance of KNN (controlled by the number of neighbors $n$) and TempCont (controlled by the chain length $m$) on eight representative tasks. (1) For KNN, as $n$ increases,

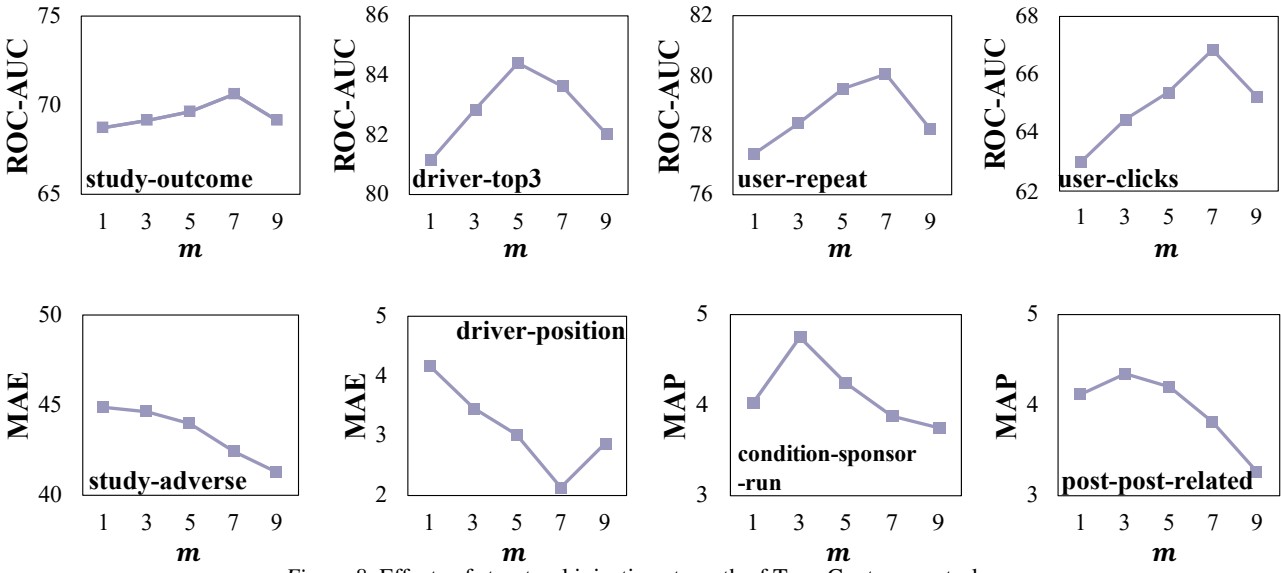

*Figure 8.* Effects of structural injection strength of TempCont across tasks.

*Table 9.* Performance on large-scale datasets across three task types. For classification we report ROC-AUC (↑), for regression MAE (↓), and for recommendation MAP (↑). Best scores are in **bold**, second best in *italics*.

| Method | user-churn (↑) | item-churn (↑) | user-engagement (↑) | user-badge (↑) | item-ltv (↓) | post-votes (↓) | item-sales (↓) | user-item-purchase (↑) | user-item-review (↑) |
|---|---|---|---|---|---|---|---|---|---|
| Base | $68.24_{\pm 0.18}$ | $80.87_{\pm 0.21}$ | $88.86_{\pm 0.16}$ | $86.83_{\pm 0.35}$ | $48.75_{\pm 0.03}$ | $0.074_{\pm 0.02}$ | $0.058_{\pm 0.02}$ | $2.64_{\pm 0.11}$ | $0.47_{\pm 0.08}$ |
| LightGBM | $65.33_{\pm 0.20}$ | $75.46_{\pm 0.25}$ | $64.94_{\pm 0.36}$ | $65.71_{\pm 0.33}$ | $53.20_{\pm 0.04}$ | $0.078_{\pm 0.02}$ | $0.079_{\pm 0.03}$ | $0.68_{\pm 0.04}$ | $0.36_{\pm 0.10}$ |
| HAN | $67.65_{\pm 0.17}$ | $78.27_{\pm 0.20}$ | $73.72_{\pm 0.30}$ | $75.57_{\pm 0.19}$ | $47.65_{\pm 0.03}$ | $0.078_{\pm 0.02}$ | $0.068_{\pm 0.02}$ | $1.37_{\pm 0.11}$ | $0.51_{\pm 0.07}$ |
| REL-GNN | $70.16_{\pm 0.15}$ | $81.66_{\pm 0.19}$ | $88.26_{\pm 0.27}$ | *$88.94_{\pm 0.25}$* | $47.02_{\pm 0.03}$ | $0.065_{\pm 0.02}$ | **$0.053_{\pm 0.02}$** | $2.88_{\pm 0.10}$ | *$0.62_{\pm 0.06}$* |
| LLM-Struct | **$73.50_{\pm 0.16}$** | *$82.51_{\pm 0.21}$* | *$90.21_{\pm 0.35}$* | $87.31_{\pm 0.54}$ | *$46.31_{\pm 0.03}$* | **$0.063_{\pm 0.02}$** | $0.056_{\pm 0.03}$ | $3.12_{\pm 0.10}$ | $0.57_{\pm 0.07}$ |
| Ours | *$72.18_{\pm 0.14}$* | **$84.17_{\pm 0.18}$** | **$91.62_{\pm 0.14}$** | **$90.47_{\pm 0.28}$** | **$44.93_{\pm 0.03}$** | $0.065_{\pm 0.02}$ | *$0.054_{\pm 0.02}$* | **$3.36_{\pm 0.09}$** | **$0.68_{\pm 0.06}$** |

performance first improves and then declines. Moderate neighborhoods yield the best results, indicating that reinforcing localized similarity is helpful, whereas larger neighborhoods introduce unrelated nodes and weaken the signal. (2) For TempCont, performance improves as $m$ increases by reinforcing short-to-medium temporal dependencies, but gains plateau at larger values. This indicates that the useful temporal horizon is limited. Extending chains too far adds little new information. Overall, these findings reinforce the principle that effective structural optimization requires task-aware calibration of both the strategy and its intensity, rather than applying fixed heuristics.

## E. Overall Predictive Performance on Large-scale Datasets

Table 9 reports the predictive performance on large-scale datasets spanning classification, regression, and recommendation tasks. Overall, our method demonstrates strong and stable performance across task types, achieving the best results on four out of six tasks and remaining competitive on the rest. On classification and regression tasks, our method consistently outperforms architectural baselines and achieves the best or second-best performance, indicating that principled structural optimization remains effective at scale. In particular, the gains on item-churn and item-ltv suggest that filtering and task-aligned structural injection help control redundancy and preserve predictive signals in large, heterogeneous graphs. For recommendation tasks (user-item-purchase and user-item-review), our method achieves the best performance, outperforming both relational GNN baselines and LLM-based structural heuristics. While LLM-Struct performs competitively on certain tasks such as user-churn and post-votes, its performance varies across tasks, whereas our approach maintains more consistent gains. Overall, these results demonstrate that the proposed structural optimization framework scales effectively to large datasets, providing robust improvements across diverse tasks without relying on task-specific heuristics or dataset-dependent tuning.

## F. Ablation Study

To better understand the contribution of each component in our structural optimizer, we conduct an ablation study on six representative RELBench tasks. We compare the full model (Ours) with several variants: Ours_ncf (**n**o **c**olumn-level **f**iltering), Ours_ntf (**n**o **t**ype-level **f**iltering), Ours_nf (**n**o **f**iltering), Ours_nv (**n**o **V**IB), Ours_ni (**n**o **i**njection).

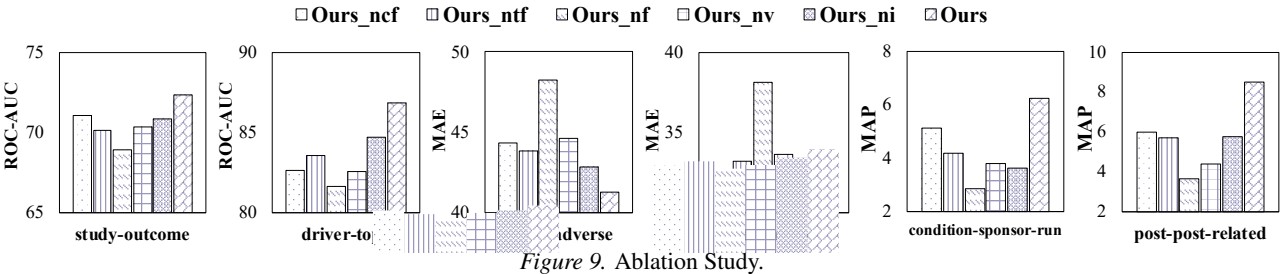

*Figure 9.* Ablation Study.

Figure 9 reports the results. We observe three clear trends from Figure 9. First, the full model consistently achieves the best performance across all six tasks, indicating that filtering, VIB, and structural injection are all useful and complementary. Removing all filtering (Ours_nf) leads to the largest degradation, especially on the regression-style tasks, showing that column/type sparsification is crucial for preventing noisy relations from dominating the graph. Second, turning off either column-level or type-level filtering (Ours_ncf and Ours_ntf) also hurts performance, but less severely than removing filtering entirely, which suggests that both levels contribute and their combination is most effective. Third, disabling injection (Ours_ni) or the VIB term (Ours_nv) consistently reduces performance compared to the full model: injection brings larger gains on tasks that benefit from additional relational paths, while VIB provides a smaller but steady improvement by regularizing node representations. Overall, these trends support our design choice that effective relational learning requires both information filtering and carefully controlled structural augmentation.

## G. Choice of Structural Complexity Metric

We compare $\ell_0$ sparsity against two alternatives: $\ell_1$ penalty and edge density penalty (Yun et al., 2019).

| Complexity Term | study-outcome (↑) | driver-top3 (↑) | condition-sponsor-run (↑) |
|---|---|---|---|
| Density penalty | 70.46 | 83.91 | 4.22 |
| $\ell_1$ penalty | 71.58 | 84.25 | 5.36 |
| $\ell_0$ (Ours) | **72.35** | **86.82** | **6.24** |

*Table 10.* Ablation on structural complexity metrics.

Table 10 reports the performance, from the table, we observe: (1) Density penalties cannot distinguish between removing one irrelevant 10,000-node table vs. ten relevant 100-node tables—both reduce edge count equally but have drastically different effects on signal quality. (2) $\ell_1$ over-smooths: it encourages all gates toward zero uniformly, whereas $\ell_0$ via Hard-Concrete creates discrete keep/drop decisions aligned with Theorem 3.2's entropy bound.

## H. Additional Theoretical Analysis and Proofs

### H.1. Proof of Theorem 3.2

Let $M = (M_1, \ldots, M_J)$ be Bernoulli gates with $\pi_j = \mathbb{P}(M_j = 1)$ and $\mu = \sum_{j=1}^{J} \pi_j = \mathbb{E}\|M\|_0$. By subadditivity of Shannon entropy,

$$H(M) = H(M_1, \ldots, M_J) \leq \sum_{j=1}^{J} H(M_j). \tag{22}$$

Since each $M_j$ is Bernoulli, $H(M_j) = h(\pi_j)$ where $h(\cdot)$ is binary entropy, hence

$$H(M) \leq \sum_{j=1}^{J} h(\pi_j). \tag{23}$$

By concavity of $h(\cdot)$ and Jensen's inequality,

$$\frac{1}{J} \sum_{j=1}^{J} h(\pi_j) \leq h\left(\frac{1}{J} \sum_{j=1}^{J} \pi_j\right) = h\left(\frac{\mu}{J}\right), \tag{24}$$

which implies

$$\sum_{j=1}^{J} h(\pi_j) \leq J\, h\left(\frac{\mu}{J}\right). \tag{25}$$

Combining yields $H(M) \leq \sum_{j=1}^{J} h(\pi_j) \leq J\, h(\mu/J)$. Finally, $h(p)$ is monotone increasing on $p \in [0, 1/2]$, so in the sparse regime $\mu/J \leq 1/2$, the bound $J\, h(\mu/J)$ increases with $\mu$. $\qquad\square$

## H.2. Structural injection as search-space enrichment

We formalize the "enrichment" claim used in Section 3.3. Recall that $R(\phi_g, \theta)$, $\Omega(\phi_g)$, and $R^\star(\phi_g)$ are defined in Eq. (10), Eq. (11), and Eq. (12), respectively. For any feasible set $\Phi$ of structural parameters, define the best achievable regularized objective

$$J^\star(\Phi) := \inf_{\phi_g \in \Phi} \left\{ R^\star(\phi_g) + \Omega(\phi_g) \right\}. \tag{26}$$

**Feasible families.**   Let $\Phi_{\text{filter}}$ denote the structural family obtainable using filtering alone (e.g., all template gates fixed to zero), and let $\Phi_{\text{full}}$ denote the family when both filtering and (gated) injection are allowed. By construction,

$$\Phi_{\text{filter}} \subseteq \Phi_{\text{full}}. \tag{27}$$

**Proposition H.1** (Monotonicity under structural enrichment). *Assume* (27). *Then*

$$J^\star(\Phi_{\text{full}}) \leq J^\star(\Phi_{\text{filter}}). \tag{28}$$

*Moreover, if there exists some $\bar{\phi}_g \in \Phi_{full} \setminus \Phi_{filter}$ such that $R^\star(\bar{\phi}_g) + \Omega(\bar{\phi}_g) < J^\star(\Phi_{filter})$, then the inequality in (28) is strict.*

*Proof.* By (27), taking the infimum over the larger feasible set cannot increase the optimum:

$$J^\star(\Phi_{\text{full}}) = \inf_{\phi_g \in \Phi_{\text{full}}} \left\{ R^\star(\phi_g) + \Omega(\phi_g) \right\} \leq \inf_{\phi_g \in \Phi_{\text{filter}}} \left\{ R^\star(\phi_g) + \Omega(\phi_g) \right\} = J^\star(\Phi_{\text{filter}}).$$

If there exists $\bar{\phi}_g \in \Phi_{\text{full}} \setminus \Phi_{\text{filter}}$ with $R^\star(\bar{\phi}_g) + \Omega(\bar{\phi}_g) < J^\star(\Phi_{\text{filter}})$, then $\bar{\phi}_g$ achieves a strictly smaller objective value than any $\phi_g \in \Phi_{\text{filter}}$, hence $J^\star(\Phi_{\text{full}}) < J^\star(\Phi_{\text{filter}})$. $\qquad\square$

