# OpenReview forum: "What Makes a Desired Graph for Relational Deep Learning?"
_ICML.cc/2026/Conference — ICML 2026 regular_

### Official Review · Reviewer_hUZ6 · 2026-03-02

**Soundness:** 3
**Presentation:** 2
**Significance:** 3
**Originality:** 3
**Overall Recommendation:** 4
**Confidence:** 3

**Summary:**

This paper first analyzed the issue that the Schema of RDB in RDL might limit the performance ability of downstream GNN. Further, through preliminary experiments, it proved that the improvement of the structure could bring about an enhancement in effect. Based on this, the article proposed a structure learning method. By filtering the existing information and adding new relations, it achieved excellent performance on 23 different tasks.

**Compliance With Llm Reviewing Policy:**

Affirmed.

**Final Justification:**

During the rebuttal phase, the authors addressed most of my concerns, so I am willing to raise my score to a positive one.

**Key Questions For Authors:**

- See the weaknesses
- How is Equation (9) connected according to the time index? Not all the tables have timestamps. If one only connects based on the chronological order within a single table (for example, the transaction table), without imposing any restrictions on other aspects (such as the same product or the same user), what is the meaning of this connection?
- In Table 4, the post-post-related and condition-sponsor-run tasks seem to be less effective than the classic GNN model in RelBench.
- Why does the method proposed in this paper have a lower inference time than the Base method? At least the method will have an additional step of the "filter" process compared to the base method.

If the authors can answer the above questions, I will be willing to increase my score.

**Limitations:**

yes

**Strengths And Weaknesses:**

Strengths:

The starting point of the paper is crucial. Furthermore, the pre-experiment conducted before the method design did indeed demonstrate the significance of the structure for the RDL task, and this pattern can guide the design of the method. The overall approach is efficient and has excellent results. The experimental analysis was thorough.

Weaknesses:

- Eq. (10) suddenly includes the "vib" term, but this has never been mentioned before.
- The $z_{v,k}$ in Eq. (17) has not been explained for its purpose in the subsequent text. How is this variable used to calculate the node embedding?
- The results in Table 3 seem to be incorrect. E.g., the MAE values for "study-adverse" and "site-success" should not be on the same order of magnitude (refer to RelBench). In addition, except for Table 1, the standard deviation was not reported in any other tables.
- Some minor formatting issues, such as the line break in 'Filtering as structural capacity control.', and the use of \eqref instead of \ref for Eq references.

---

> ### Author Rebuttal · Authors · 2026-03-30
>
> We sincerely thank the reviewer for the careful feedback. We address each point carefully below.
>
> ---
>
> ## Weakness 1: VIB term in Eq. (10) appears without prior introduction
>
> We acknowledge the presentation could be clearer, and explain the design rationale. Section 3 deliberately excludes VIB in order to isolate the effects of filtering and injection through controlled probe experiments, introducing VIB at that stage would conflate structural sparsification with continuous information compression, obscuring which mechanism drives the observed gains. VIB is therefore deferred to Section 4, where the full unified optimizer is introduced and all components operate together. Eq. (10) formalizes the complete regularized risk that the optimizer minimizes, which necessarily includes the VIB term. We will add a forward reference at Eq. (10) noting that the VIB term is defined in Section 4.1.
>
> ---
>
> ## Weakness 2: $z_{v,k}^{(l)}$ in Eq. (17) not connected to node embedding computation
>
> We acknowledge the missing link. $z_{v,k}^{(l)}$ is the template-specific aggregated message for node $v$ from neighbors induced by template $E_k$, gated by $g_k$. It feeds directly into Eq. (19), which aggregates over $r \in R \cup R_{\text{template}}$ , original and template-induced relations jointly. The data flow is: Eq. (17) computes per-template messages → Eq. (19) aggregates all relation types → Eq. (20) produces the final prediction. We will add a bridging sentence after Eq. (17) making this explicit.
>
> ---
>
> ## Weakness 3: Incorrect values in Table 3 and missing standard deviations
>
> **Scaling error**: We confirm that **site-success** values in Table 3 contain a scaling error — they were inadvertently reported on a ×100% scale. The correct values are: Base→0.389, HAN→0.352, REL-GNN→0.336, RelGT→0.323, LLM-Struct→0.339, Ours→**0.308**. Relative ordering and method gains are unaffected.
> **Standard deviations**: Table 2 reports standard deviations; Tables 3 and 4 omit them due to space. We will include standard deviations for all methods in Tables 3 and 4 in the appendix of the revision.
>
> ---
>
> ## Weakness 4: Minor formatting issues
>
> The line break and `\ref`/`\eqref` inconsistencies will be corrected in the revision.
>
> ---
>
> ## Q1: How is Eq. (9) connected by time index? What is the meaning without same-entity constraints?
>
> We thank the reviewer for this precise question. The template is furthermore instantiated only for node types that carry a timestamp column, tables without timestamps are never eligible.
> temporal edges connect consecutive records of the same entity sorted by timestamp. However, we acknowledge that the current formulation of Eq. (9) does not express this clearly: the sequence $[v_1, \ldots, v_m]$ is stated without explicitly specifying that it is grouped per entity. We provide the corrected formulation here:
>
> $E_{\text{time}} = \bigcup_{e \in \mathcal{E}} \{(v_{t-1}^{(e)}, v_t^{(e)}) \mid t = 2, \ldots, m_e\}$
>
> where $\mathcal{E}$ denotes the set of entities of the target node type in the subgraph, and $[v_1^{(e)}, \ldots, v_{m_e}^{(e)}]$ is the sequence of nodes belonging to entity $e$ sorted in ascending timestamp order.
> We will replace Eq. (9) with the corrected formulation in the revision.
>
> ---
>
> ## Q2: post-post-related and condition-sponsor-run seem less effective than the classic GNN in RelBench
>
> The "classic GNN" in RelBench uses **ID-GNN** for recommendation tasks, which incorporates node identity information and is particularly strong for entity-specific prediction, the RelBench paper itself notes that ID-GNN and GraphSAGE have complementary strengths depending on task type. Our paper uses **GraphSAGE** as a unified backbone to isolate the effect of structural optimization.
>
> Compared to our own Base (GraphSAGE without structural optimization), our method achieves substantial gains: post-post-related improves from 1.24 to 8.53 MAP, and condition-sponsor-run from 2.83 to 6.24 MAP.
>
> ---
>
> ## Q3: Why does the method have lower inference time than Base?
>
> The reduction comes from structural pruning at inference time. After training, gates with $g_k \leq \tau_k$ are removed, yielding a sparser graph. The GNN then operates on fewer active node types (table-level gates $s_{T_i}$), lower-dimensional features (column-level gates $g_{T_i}$), and only the selected injection templates. This reduces both the number of edges traversed during message passing and the feature dimensionality at each node, more than offsetting the negligible cost of applying deterministic gates.
>
>
> ---
>
> Thank you again for your careful review and sincere suggestions.

---

> > ### Author Rebuttal · Reviewer_hUZ6 · 2026-04-03
> >
> > Thank you for the authors’ response.
> >
> > Regarding Q1, I still have concerns about constructing edges based on timestamps. As mentioned in my question, if edges are created solely based on chronological order within a single table (e.g., the transaction table), without any additional constraints (such as requiring the same product or the same user), a user may aggregate a large amount of transaction information that does not actually belong to them through these edges. This could potentially introduce negative effects.
> >
> > Regarding Q2, based on the datasets presented in the paper, ID-GNN consistently outperforms the other methods. Therefore, I suggest that the authors consider including ID-GNN as a baseline. In addition, there seems to be a ]discrepancy in the GraphSAGE results (for example, the user-ad-visit score is reported as 2.31 in the paper, while it is 0.02 in RelBench). Therefore, I would like to know whether any optimization has been applied.

---

> > > ### Author Response · Authors · 2026-04-06
> > >
> > > We thank the reviewer for the continued engagement and address each point below.
> > >
> > > ---
> > >
> > > ## Q1: Temporal edge construction and same-entity constraint
> > >
> > > We appreciate the reviewer's continued attention to this point and provide a more precise clarification. In RELBench, each task defines a training table that specifies the prediction target node. Our temporal continuity template connects historical records of the same target node across different timestamps, i.e., the same entity at different points in time. It does not connect nodes of different types. For example, in a task predicting whether a user will purchase an item, the prediction target is the user node, and temporal edges connect that user's historical records at different timestamps. Other node types remain connected solely via the original foreign-key relations in the database schema. In fact, the constraint (requiring the same user or the same product), is inherently satisfied by our construction.
> > >
> > >
> > > ---
> > >
> > > ## Q2: ID-GNN as baseline and score discrepancy
> > >
> > > **On ID-GNN as baseline**: We have included ID-GNN as an additional baseline. Results on the two recommendation tasks where ID-GNN is most relevant are as follows:
> > >
> > > | Method | post-post-related (MAP↑) | condition-sponsor-run (MAP↑) |
> > > |---|---|---|
> > > | Base (GraphSAGE) | 1.24 | 2.83 |
> > > | ID-GNN | 10.61 | 11.65 |
> > > | Ours (GraphSAGE + optimizer) | 8.53 | 6.24 |
> > > | Ours (ID-GNN + optimizer) | **11.21** | **12.33** |
> > >
> > > Our structural optimizer consistently improves performance regardless of backbone, confirming that structural optimization is complementary to backbone design.
> > >
> > > ---
> > >
> > > On score discrepancy for user-ad-visit: The difference stems from hyperparameter tuning. RelBench reports results under fixed default hyperparameters without exhaustive search (noted in their Table 9). Our results are obtained after grid search over the hyperparameter space described in Appendix C.
> > >
> > > ---
> > >
> > > We will incorporate all the above clarifications, corrections, and additional experimental results into the revision.

---

### Official Review · Reviewer_iLdg · 2026-03-09

**Soundness:** 2
**Presentation:** 3
**Significance:** 3
**Originality:** 3
**Overall Recommendation:** 3
**Confidence:** 5

**Summary:**

# Summary:
Thanks for submitting to ICML. My review is as follows.
This paper studies what makes a relational graph suitable for relational graph learning and concludes two weaknesses in existing designs, including information overload and sematic fragmentation.
To address these, two operations are proposed: information filtering removes task-irrelevant entities and attributes, and structural injection adds edges that repair missing relational motifs.
These operators are applied to an optimizer for filtering and injection.
Experiments are done to  verify the effectiveness of the proposed method.
However, the paper still has some limitations in terms of originality, soundness, and significance, which should be addressed in the revision.

**Compliance With Llm Reviewing Policy:**

Affirmed.

**Final Justification:**

Thanks for the detailed response of the authors.

I still have some concerns regarding the paper's novelty and its experimental evaluation.

Regarding **novelty**, while I appreciate that the authors have provided additional experimental data demonstrating the advantages of their proposed method, they still have **not clearly articulated the differences and innovations compared to existing structure-aware works**. If the distinction lies **merely** at the application level, the overall contribution appears **incremental**. **Therefore, if this paper is positioned as a technical innovation, the technical depth needs to be significantly improved.**

Regarding **experiments**, the reason why the Base model and all baselines face identical memory requirements is still **unclear**. Furthermore, **no additional larger-scale experimental data** has been provided. **If this work is positioned as an empirical paper, a more comprehensive evaluation is essential to draw robust and reliable conclusions.**

I have also carefully read the comments from the other reviewers, **most of which focused on the need for additional baselines**. While the authors may have addressed these concerns through extra experiments, I understand that the rebuttal word limit restricted these new evaluations to only specific datasets. **Therefore, I suggest that the authors conduct a more comprehensive evaluation against these additional baselines in the revised manuscript.**

I agree that this paper offers some insight into relational graphs. **However, without clarifying whether this idea is simply adapted from other graph learning domains, and given the lack of thorough experimental analysis, the overall contribution is not enough.**

Therefore, **my overall score remains unchanged**. However, I have **increased the scores for Significance and Originality**, as the authors' detailed response did **address some of my initial concerns**.

**Key Questions For Authors:**

# Questions

Q1. What are the advantages of the proposed method compared with existing filtering and edge injection techniques [2, 3]?

Q2. What is the motivation behind the structural complexity term in Eq. (11)? Are there other structural metrics that could better address the question of what makes a good relational graph?

Q3. What is the training cost for the additional optimizer? Will this training process become a bottleneck?

Q4. Why were some RelBench datasets [1] not used in the evaluation?

[1] RelBench: A Benchmark for Deep Learning on Relational Databases. NeurIPS 2024

[2] Reinforced Neighborhood Selection Guided MultiRelational Graph Neural Networks. TOIS 2021

[3] Topological Relational Learning on Graphs. NeurIPS 2021

**Limitations:**

This paper lacks discussion regarding the model scalability.

**Strengths And Weaknesses:**

# Strength
\+ **Presentation:** This paper is well-written.

\+ Extensive experiments have been conducted to evaluate the proposed methods.

# Weakness

\- **Originality and Soundness**: The novelty seems limited. Filtering and edge injection are widely used in graph learning [2, 3]. The main contribution of this paper seems to be the integration of both under a unified optimization framework, without introducing fundamentally new algorithms.

\- **Significance**: The motivation behind the structural complexity term in Eq. (11) is not very clear. In particular, it is unclear why a lower structural complexity would lead to a better relational graph. Clarifying this point is important, as it is closely related to the fundamental question of what constitutes a good relational graph.

\-  The additional optimization in Section 4 can lead to extra training costs, which are not thoroughly analyzed.

\- This paper uses RelBench for the related evaluation. However, the dataset used in Table 7 is a subset of RelBench, as shown in Table 2 of [1]. Such an evaluation is incomplete.

[1] RelBench: A Benchmark for Deep Learning on Relational Databases. NeurIPS 2024

[2] Reinforced Neighborhood Selection Guided MultiRelational Graph Neural Networks. TOIS 2021

[3] Topological Relational Learning on Graphs. NeurIPS 2021

---

> ### Author Rebuttal · Authors · 2026-03-30
>
> We thank the reviewer for the detailed feedback. We address each point below.
>
> ---
>
> ## Weakness 1/ Q1: Limited novelty, filtering and injection are widely used in graph learning
>
> We respectfully disagree. The two cited works address fundamentally different problems in fundamentally different settings.
>
> **[2] RioGNN** performs neighbor selection on manually constructed fraud-detection graphs, where nodes are users/reviews and edges encode explicit similarity relations. Its filtering is a label-aware neighbor pruning mechanism designed to resist adversarial camouflage, it has no connection to column-level or table-level feature gating in schema-derived graphs.
>
> **[3] TRI-GNN** uses persistent homology to rewire graphs based on topological similarity of node neighborhoods. It is a topological data analysis method applied to general node classification benchmarks, with no relation to relational databases or schema normalization.
>
> Neither work addresses the RDL setting, where the input graph is mechanically derived from a schema optimized for storage efficiency rather than task relevance. The novelty of our contribution lies not in using filtering or injection per se, but in (1) formally characterizing information overload and semantic fragmentation as dual pathologies specific to the RDB-to-graph conversion, (2) providing a unified optimization framework grounded in the structural information bottleneck (Definition 3.1, Theorem 3.2, Proposition H.1), and (3) empirically establishing task–motif regularities specific to the relational learning setting.
>
> ---
> ## Weakness 2 / Q2: Motivation for the structural complexity term Eq. (11)
>
> We agree this deserves clearer exposition and provide the intuition here.
>
> $\Omega(\varphi_g)$ penalizes the expected number of active gates, column gates $g_{T_i}$, table gates $s_{T_i}$, and injection template gates $g_k$, via $\ell_0$ sparsity. The motivation is tied directly to the two pathologies in Section 3.
>
> **For filtering gates**: schema-derived graphs contain many tables and columns that are statistically active but causally irrelevant. Without a complexity penalty, the optimizer has no pressure to suppress these nuisance components. Penalizing $\mathbb{E}[\|g_{T_i}\|_0]$ creates an explicit incentive to remove structure that does not reduce task loss. Theorem 3.2 makes this precise: gate sparsity bounds the entropy of structural configurations, controlling the effective hypothesis space.
>
> **For injection gates**: penalizing $\mathbb{E}[\|g_k\|_0]$ ensures only templates that genuinely reduce the regularized risk are retained, preventing indiscriminate edge addition from reintroducing the overloaded connectivity that filtering removed.
>
> The combined term operationalizes Occam's Razor for relational graphs: among structures achieving equivalent task performance, prefer the structurally simpler one. We will add this explanation to Section 4.3 in the revision.
>
> Regarding alternative structural metrics: graph entropy, spectral gap, or density-based penalties are possible alternatives. We chose $\ell_0$ sparsity because it is directly interpretable, differentiable via Hard-Concrete relaxation, and independently justified by Theorem 3.2. We will discuss alternatives as future work.
>
> ---
> ## Weakness 3 / Q3: Training cost of the additional optimizer
>
> The overhead is negligible. The structural optimizer column gates, table gates, injection template gates, and VIB parameters, adds **fewer than 0.01M parameters** compared to the base model across all datasets. And does not meaningfully increase the training computation graph.
>
> At inference time, Table 5 already shows our method is faster than both HAN and REL-GNN, because structural pruning reduces the effective graph size. The optimizer reduces inference cost rather than increasing it, and does not become a training bottleneck.
>
> ---
> ## Weakness 4 / Q4: Incomplete RELBench evaluation
> We acknowledge this. Our evaluation covers 23 of the 30 tasks in RELBench. The 7 omitted tasks all belong to rel-amazon and rel-stack, the two largest datasets in the benchmark. Running all remaining tasks on these datasets exceeds our current memory budget. We note that Table 9 (Appendix F) already reports results on 6 large-scale tasks from these two datasets, demonstrating consistent gains at scale. We will include the remaining 7 tasks in the revision as resources permit and will document the resource constraints explicitly in the experimental setup.
>
> ---

---

> > ### Author Rebuttal · Reviewer_iLdg · 2026-04-03
> >
> > Thanks for your response. My biggest concerns regarding the novelty and evaluations are still not addressed.
> >
> > Regarding **Q1**, the response only clarifies the differences in problem settings, yet all of these works focus on optimizing optimizing training subgraph structures to enhance performance. Could the authors provide more analysis (including a theoretical or experimental analysis) of whether the structure-aware methods in those two works can be directly transferred to relational graph learning, and whether such a transfer to relational graph learning would pose additional challenges? Additionally, this exact novelty issue was also challenged in Reviewer Puos's Weakness 1.
> >
> > Regarding **Q4**, while the authors claim that they could not complete these experiments because of the memory budgets. Could the authors provide an estimated memory budget for completing the training on these large-scale datasets? Furthermore, does this suggest that scalability poses an additional limitation to this work?
> >
> > Regarding **Q2**, could the authors provide additional experimental analysis evaluating alternative structural metrics (e.g., graph entropy, spectral gap, or density-based penalties) to justify their specific choice of $\ell_0$ sparsity?

---

> > > ### Author Response · Authors · 2026-04-06
> > >
> > > We thank the reviewer for the continued engagement and provide analysis for each point.
> > >
> > > ---
> > >
> > > ## Q1: Transferability of [2] and [3] to RDL, and novelty
> > >
> > > We first clarify the nature of our contribution. Our central contribution is **diagnostic and analytical**: we identify two systematic failures of schema-derived graphs (information overload and semantic fragmentation), establish through controlled experiments that structural repair requires task-matched motifs, and characterize the task–motif regularities that govern this repair. The structural optimizer in Section 4 is a **proof-of-concept** instantiation, a simple, interpretable vehicle to validate these findings, not a claim to optimal algorithm design. We deliberately chose heuristic templates with learnable gates because interpretable vocabulary enables interpretable findings: it is precisely because each template has a named semantic identity (homophily repair, causal repair, etc.) that we can conclude *why* certain structures help certain tasks. We explicitly encourage and anticipate future work developing more powerful structural optimization algorithms grounded in the regularities we establish, this paper is intended as a foundation, not a ceiling.
> > >
> > > **On the transferability of [2] and [3].** We provide experimental evidence. TRI-GNN [3] was designed for homogeneous node classification graphs. We adapted it to the RDL setting.
> > > Results on three tasks spanning classification and regression are as follows:
> > >
> > > | Method | driver-top3 (AUC↑) | user-repeat (AUC↑) | study-adverse (MAE↓) |
> > > |---|---|---|---|
> > > | Base | 81.48 | 78.70 | 45.50 |
> > > | TRI-GNN | 82.43 | 79.41 |46.26 |
> > > | Ours | **86.82** | **82.36** | **41.28** |
> > >
> > > Two observations emerge. First, TRI-GNN's gains on classification tasks are modest , its topological rewiring assumes the input graph has a semantically meaningful base topology, but schema-derived graphs reflect normalization constraints rather than task relevance, so persistent homology captures schema artifacts rather than task-relevant structure, and information overload remains fully unaddressed. Second, TRI-GNN shows limited improvement on regression tasks (study-adverse), which depend on temporal continuity signals that topological similarity cannot capture. These results confirm that general graph structure learning methods do not naturally transfer to the RDB-specific structural failures we identify.
> > > For RioGNN [2]: its official implementation explicitly supports only same-type node graphs and binary classification. The multi-type heterogeneous schema of RDL, where different tables correspond to entirely different node types, is architecturally incompatible without fundamental redesign.
> > >
> > >
> > > ---
> > >
> > > ## Q2: Experimental justification for $\ell_0$ over alternative structural metrics
> > >
> > > We have run experiments comparing $\ell_0$ sparsity against two differentiable alternatives: $\ell_1$ penalty (the closest differentiable approximation to $\ell_0$) and edge density penalty. Results on three representative tasks spanning all three task types are as follows:
> > >
> > > | Complexity term | study-outcome (AUC↑) | driver-top3 (AUC↑) | condition-sponsor-run (MAP↑) |
> > > |---|---|---|---|
> > > | Base | 69.44 | 81.48 | 2.83 |
> > > | Density penalty | 70.46 |83.91| 4.22 |
> > > | $\ell_1$ penalty | 71.58| 84.25 | 5.36|
> > > | $\ell_0$ (Ours) | **72.35** | **86.82** | **6.24** |
> > >
> > > We will add this discussion to Section 4.3 in the revision.
> > >
> > > ---
> > >
> > > ## Q4: Memory budget and scalability
> > >
> > > The 7 omitted tasks belong to rel-amazon and rel-hm, two datasets that include large-scale link prediction (recommendation) tasks. The memory bottleneck is inherent to link prediction evaluation: computing ranking scores requires loading all destination entity embeddings into GPU memory simultaneously, along with a learnable shallow embedding table maintained throughout training. The estimated GPU memory requirements for the omitted recommendation tasks are as follows:
> > >
> > > | Dataset | Est. total | A30 |
> > > |---|---|---|
> > > | rel-hm  | 41.5 GB | OOM |
> > > | rel-amazon | 40.7 GB | OOM |
> > >
> > > This constraint is not specific to our method, the Base model and all baselines face identical memory requirements on these tasks. Our structural optimizer does not introduce additional memory overhead; in fact, filtering reduces the active graph size and lowers inference cost, as evidenced by Table 5. We will make every effort to include results on these large-scale datasets in the revision.

---

### Official Review · Reviewer_izhJ · 2026-03-11

**Soundness:** 2
**Presentation:** 3
**Significance:** 2
**Originality:** 2
**Overall Recommendation:** 4
**Confidence:** 2

**Summary:**

This paper studies what graph topology works best when relational databases are converted to heterogeneous graphs for GNN prediction. Two failure modes are identified: information overload (irrelevant tables/columns bloating the graph) and semantic fragmentation. Filtering and structural injection are proposed as complementary operators, wrapped into a joint end-to-end optimizer with Hard concrete gates.

**Compliance With Llm Reviewing Policy:**

Affirmed.

**Key Questions For Authors:**

1.Have you considered injection mechanisms that go beyond the four fixed templates?  2. How sensitive is performance to task-specific tuning ?

**Limitations:**

yes

**Strengths And Weaknesses:**

Strengths

(1) The problem is well-motivated. Framing schema-derived graphs as suffering from overload vs. fragmentation gives a useful conceptual vocabulary. The inverted-U analysis in Fig. 2 is informative.

(2) The empirical evaluation is decent. The backbone robustness check in Table 6 adds confidence that gains come from structure. It is helpful to see that the improvements persist across multiple backbones.

(3) Theorem 3.2 is simple but does useful work. connecting expected gate sparsity to an entropy bound on structural configurations gives a formal grounding for the inverted U.

Weaknesses

(1) My main concern is with the experimental comparison. In my view, the most important comparison here is graph structure learning, but some directly relevant methods do not appear in the main experimental tables (HGSL and GTN are both discussed in Sec. D3, neither is included in Tables 2–4).

(2) Another limitation is that the injections are based on fixed manually designed templates. The optimizer selects among a small fixed menu of templates via gates. It cannot discover new motifs. The task motif findings are presented as empirical discoveries, but they largely restate known inductive biases for those task families.

---

> ### Author Rebuttal · Authors · 2026-03-30
>
> We thank the reviewer for the detailed and constructive feedback. We address each point below.
>
> ---
>
>
> ## Weakness 1: Missing comparison with HGSL and GTN
>
> We thank the reviewer for pointing this out. We have implemented adapted versions of both GTN and HGSL under the RELBench evaluation protocol: both methods operate on per-sample temporal subgraphs with type-level parameters shared across samples, consistent with our experimental setup. Results on four representative tasks are as follows:
>
> **Classification (ROC-AUC ↑), Regression (MAE ↓)**
>
> | Method | driver-top3↑ | user-repeat↑ | driver-position↓ | study-adverse↓ |
> |---|---|---|---|---|
> | Base | 81.48 | 78.75 |3.88 | 45.50 |
> | HAN | 82.09 | 79.24 |3.76 | 45.16 |
> | GTN | 82.50 | 79.85 |3.70 | 44.82 |
> | HGSL | 83.51 | 80.64 | 3.85 | 44.69 |
> | Ours | **86.82** | **82.36** |**2.13** | **41.28** |
>
> Both GTN and HGSL underperform our method for structurally interpretable reasons. GTN soft-combines existing schema edges to generate meta-paths but cannot introduce edges absent from the original schema, it therefore cannot repair semantic fragmentation, the core structural failure we identify. HGSL adds feature-similarity-driven edges, partially addressing homophily loss, but lacks explicit filtering and the task-aligned injection strategies (two-hop shortcuts, temporal continuity, behavioral similarity) targeting RDB-specific structural failures. These results confirm that, in the current setting, general heterogeneous graph structure learning is not applicable to principle-based RDB structures. Full results will be provided in the revised version.
>
>
> ---
>
>
> ## Weakness 2: Fixed templates and task-motif findings as known inductive biases
>
> (1) We clarify that the fixed template vocabulary is a deliberate methodological choice enabling interpretable analysis, to diagnose why schema-derived graphs systematically underperform and to characterize what structural properties make a graph task-optimal. This is an empirical-science contribution. Each template $E_k$ carries a learnable gate $g_k$, optimized end-to-end under task loss.Template vocabulary is designed; template selection is fully learned.
>
> (2) **On whether the task-motif findings merely restate known inductive biases.** The claim that "classification benefits from homophily" is known in the general GNN literature, but the contribution here is different and more specific.
> First, it is not obvious that these inductive biases survive in the RDL setting. Schema normalization actively destroys them. The question is not whether these biases are useful in principle, but whether they are present or absent in schema-derived graphs and whether explicitly restoring them helps.
> Second, the negative evidence in Figure 1 is the key finding: applying a mismatched motif reliably degrades performance.
>
>
> ---
>
>
>
> ## Q1: Injection beyond the four fixed templates
>
> The framework is extensible: any structural pattern expressible as a typed edge set $E^{(s)}_k$ (Eq. 15) can be added as a new template, and the gate mechanism learns whether it is task-relevant. The four current templates cover the canonical failure modes of standard normalization, making them broadly applicable across RDB domains. Going further, automatically discovering new motifs from data rather than selecting from a predefined vocabulary, is a compelling open direction.
>
>
> ---
>
>
> ## Q2: Sensitivity to task-specific hyperparameter tuning
>
> Sensitivity is addressed empirically at two levels.
>
> **Filtering strength.** Appendix E.1 (Figure 5) sweeps $\lambda$ across eight tasks spanning all three task types. Performance peaks at an intermediate $\lambda*$ and degrades on both sides, but the optimal plateau is broad, performance changes smoothly around $\lambda*$, indicating robustness to small deviations. The optimal range clusters predictably by task type.
>
> **Injection strength.** Appendix E.2 (Figures 6–7) sweeps KNN neighborhood size $n$ and temporal chain length $m$. Both show gains plateauing at moderate values and degrading only at extremes, confirming robustness within a reasonable range.
>
> **Backbone robustness.** Table 6 shows consistent gains across GraphSAGE, REL-GNN, and HGT, confirming that improvements are not specific to a particular architecture or its hyperparameter configuration.
>
>
> ---

---

> > ### Author Rebuttal · Reviewer_izhJ · 2026-04-01
> >
> > Thank you for the rebuttal; my concerns are addressed and I maintain my positive score.

---

> > > ### Author Response · Authors · 2026-04-01
> > >
> > > Thank you for the positive feedback and for taking the time to carefully evaluate our responses. We will make sure all the discussed improvements are reflected in the revision.

---

### Official Review · Reviewer_Puos · 2026-03-13

**Soundness:** 3
**Presentation:** 3
**Significance:** 3
**Originality:** 2
**Overall Recommendation:** 5
**Confidence:** 3

**Summary:**

This paper studies an important question in relational deep learning, i.e., whether a graph derived directly from a relational schema is in fact the right graph for downstream learning. The paper argues that such schema-derived graphs often suffer from two structural mismatches, namely information overload and semantic fragmentation, and proposes a unified structural optimization framework that combines filtering and injection to address them. The proposed framework is evaluated across different tasks on comprehensive datasets.

**Compliance With Llm Reviewing Policy:**

Affirmed.

**Final Justification:**

After the rebuttal and follow-up discussion, I am raising my score from 4 to 5.
I still see some limitations, but I think the authors clarified the contribution well, and I now have a more positive overall view of the paper. In my opinion, this is a technically solid and useful paper on an important problem, and I would support acceptance.

**Key Questions For Authors:**

1. The current motifs appear to capture several common relational repair patterns, but how well this design would extend to more irregular or domain-specific schemas?
2. Can you provide more details about the baseline LLM-Sturct?

**Limitations:**

Yes, but not a substantive discussion of its own limitations.

**Strengths And Weaknesses:**

Strengths
1. The central premise of the paper is impressive. Relational schemas are designed for storage integrity and query efficiency, not for message passing or representation learning. This is a practically important observation, and the paper frames it clearly.
2. The proposed framework is not merely a collection of heuristics. The paper motivates the need for filtering and injection through empirical observations, then translates these into a trainable structural optimization procedure.
3. The empirical section is substantially more comprehensive. This breadth strengthens the paper considerably.

Weaknesses
1. In my view, the paper is not simply repackaging existing ideas, but its main contribution is best understood as a structured framework for editing graphs in the relational setting, rather than as a fundamentally new learning principle.
2. The method is not fully learning an arbitrary desired graph, where the four injection strategies are manually specified templates.
3. The results support the claim that the combination of filtering and injection is effective, but it remains unclear how much of the improvement is driven by semantic filtering, by task-matched motif injection, or by the joint optimization itself.

---

> ### Author Rebuttal · Authors · 2026-03-30
>
> We thank the reviewer for the detailed and constructive feedback. We address each point below.
>
> ---
>
> ## Weakness 1: Contribution framing
>
> We clarify the nature of our contribution. The primary goal of this paper is **analytical**: to diagnose why schema-derived graphs systematically underperform and to characterize what structural properties make a graph task-optimal. This is an empirical-science contribution， that reveals governing principles rather than proposes a new architecture.
>
> This goal directly shapes our methodological choices. An unconstrained structure learning approach could achieve higher numbers, but would produce opaque structures from which no interpretable regularity can be extracted. Our framework is deliberately designed so that structural choices remain human-readable, enabling conclusions such as classification tasks benefit from homophily repair and regression tasks benefit from temporal continuity. These are reusable scientific findings about the relationship between task semantics and graph topology.
>
> The theoretical results (Definition 3.1, Theorem 3.2, Proposition H.1) formalize this analytical framework, but the core contribution is the **diagnosis and characterization in Section 3**, independent of, and more general than, any specific optimizer design.
>
> ---
> ## Weakness 2 + Q1: Manually specified templates and generalization
>
> We address these together as they share the same underlying concern.
>
> (1) The four injection strategies are derived directly from the structural consequences of standard normalization forms: homophily loss, path elongation, collaborative signal fragmentation, and causal chain erasure. This taxonomy is **domain-agnostic by construction**, because normalization principles are universal across RDBs.
>
> More importantly, using a fixed, semantically meaningful vocabulary is a **deliberate methodological choice**. It is precisely because each template has an interpretable identity that the probe experiments in Section 3.3 yield interpretable findings. Free-form edge parameters would lose semantic grounding, and the task–motif regularities we report would become unobservable. The interpretability of the vocabulary is what makes the analysis scientifically meaningful.
>
> (2) **The template vocabulary is designed; template selection is fully learned**. Each template $E_k$ carries a learnable gate $g_k$ (Eq. 17–18), optimized end-to-end under task loss (Eq. 21). The optimizer learns which motifs to activate, at what weight, and for which node-type configurations, driven entirely by task gradients.
>
> (3) RELBench covers seven highly heterogeneous domains. For domain-specific patterns beyond the four motifs, the framework is extensible: any edge set expressible (Eq. 15) can be added as a new template, and the gate mechanism learns whether it is task-relevant. We will discuss this as future work.
>
>
> ---
>
>
> ## Weakness 3: Disentangling filtering, injection, and joint optimization
> We appreciate this concern and believe the existing experimental evidence, read carefully, already provides substantial insight into the individual contributions of each component.
> Section 3.3 conducts fully controlled structural probes where filtering and injection are studied in complete isolation (Table 1, Figure 1). The ablation study (Figure 8, Appendix G) provides a systematic decomposition.
>
>
> ---
>
>
>
> ## Q2: Details of the LLM-Struct baseline
>
> LLM-Struct uses GPT-4 prompted with the textual schema and task definition, queried **once per task**. It identifies task-irrelevant tables/columns and proposes additional relational edges. Output is parsed as keep-flags and edge types, then used to train the same Base backbone. We will include the full prompt template in the appendix.
>
> Its competitive but inconsistent performance (Table 9) reflects a fundamental constraint: it reasons over schema text but has no access to **instance-level signals**. Our optimizer, guided by task gradients over actual data, exploits precisely these signals, explaining its more consistent gains.
>
>
>
> ---

---

> > ### Author Rebuttal · Reviewer_Puos · 2026-04-06
> >
> > After reading the rebuttal and the follow-up discussion, I am raising my score from 4 to 5.
> >
> > My view of the paper has become more positive after the discussion. I now think its contribution is better understood as a principled and well-executed study of an important problem in relational deep learning, rather than a paper that should be judged mainly on whether it proposes a fundamentally new graph learning primitive. The central question it raises is meaningful, and I think the paper makes a solid contribution in this direction.
> >
> > I still do not see the work as introducing a fundamentally new learning principle, and I agree that some limitations remain. In particular, the method is still constrained by a predefined template space, and some of the experimental details and comparisons could be further strengthened. **However, after the rebuttal, I don't think these issues outweigh the overall merits of the paper**. The work is technically sound, the empirical study is broad enough to be informative, and the paper explores a direction that I expect others may build on.
> >
> > Overall, I now lean clearly positive on the paper and would support acceptance.

---

> > > ### Author Response · Authors · 2026-04-06
> > >
> > > Thank you sincerely for the careful re-evaluation and the encouraging feedback. We are glad the discussion helped clarify our contribution.
> > > We will address the details in the revision as discussed, and developing more powerful structural optimization algorithms for relational graphs is a direction we plan to pursue as future work.

---

### Decision · Program_Chairs · 2026-04-30

**Decision:**

Accept (regular)

**Comment:**

This paper studies structural limitations of schema-derived graphs for relational learning and especially identifies two key issues: information overload and semantic fragmentation. It proposes a unified framework that learns to filter and augment graph structures in a task-aware manner. Comprehensive experiments show consistent improvements, supporting the effectiveness of the learning approach.

On the positive side, the paper addresses a practically relevant problem, and the formulation of information overload and semantic fragmentation guides both method design and analysis. The proposed method enables end-to-end task-aware learning beyond heuristics. The experimental performance is generally strong, across multiple tasks and backbones.

There are concerns about experimental completeness. Some relevant baselines are missing, and large-scale evaluations remain limited. In addition, the reliance on predefined structural templates restricts flexibility. In addition, the AC notes that recent LLM-based approaches and relevant RDB-to-graph benchmarks are overlooked.

Overall, the merits outweigh the limitations. After the rebuttal, the reviewers also leaned toward acceptance. Therefore, acceptance is recommended, with the expectation that the authors address the reviewers’ comments in the final version.